# The impact of college students' physical exercise on negative emotion: The chain mediating role of self-efficacy and smartphone addiction

**Rui Li**[1,2☯], **Hao-yu Li**[2☯], **Hao-jie Zuo**[2☯], **Jia-meng Sun**[1,3], **Qi Liu**[2], **Chen-xi Li**[2], **Ning Zhou**[2], **Bo Li**[2], **Mohamad Nizam Nazarudin**[1]*

**1** Faculty of Education, Universiti Kebangsaan Malaysia, Bangi, Selangor, Malaysia, **2** Institute of Sports Science, Nantong University, Nantong, China, **3** School of Physical Education, Nantong Normal College, Nantong, China

☯ These authors contributed equally to this work.
* mohdnizam@ukm.edu.my

## Abstract

### Objective

Against the backdrop of widespread smartphone adoption, rising student stress levels, and the increasing prevalence of negative emotions, this study employs a chain-mediation model to investigate the relationship between physical exercise and negative emotions among university students. It further elucidates the mediating roles of self-efficacy and smartphone addiction in this association.

### Methods

Data from the 2022 China Physical Activity and Health Longitudinal Survey (CPAHLS-CS) were used, involving a sample of 16,355 college students from East China and Central China. Measurements were taken using the Physical Activity Rating Scale (PARS-3), the Depression, Anxiety, and Stress Scale (DASS-21), the Physical Activity Self-Efficacy Scale (PASS), and the Mobile Phone Addiction Tendency Scale (MPATS). Descriptive analysis, correlation analysis, regression analysis, and chain mediation effect testing were performed using SPSS 27.0, Excel, and the PROCESS plugin, controlling for confounding variables such as gender, grade, region, and age.

### Results

The study found a significant negative correlation between physical exercise and negative emotions in university students ($r = -0.564$, $p < 0.001$). Self-efficacy and smartphone addiction both played significant mediating roles in the relationship between physical exercise and negative emotion. The mediation effects included independent mediation by self-efficacy and smartphone addiction, as well as a

**Data availability statement:** Due to ethical restrictions of disclosing personal and sensitive data in accordance with the protocol approved by the Ethics Review Board of Nantong University, authors have to seek permission to allow us to make the data used in this study available. Access can be requested from the Ethics Review Board of Nantong University, via email (kjc@ntu.edu.cn).

**Funding:** BL was supported by 2025 Major Project of Philosophy and Social Science Research in Colleges and Universities (No: 2025SJZD143).

**Competing interests:** The authors have declared that no competing interests exist.

**Abbreviations:** CPAHLS-CS, Chinese University Students Physical Activity and Health Longitudinal Survey; PARS-3, the Physical Activity Rating Scale; DASS-21, the Depression, Anxiety, and Stress Scale; PASS, the Physical Activity Self-Efficacy Scale; MPATS, the Mobile Phone Addiction Tendency Scale; SCT, Social Cognitive Theory.

chain mediation effect between the two, with effect sizes of 0.005 and −0.017, respectively.

## Conclusion

This study offers a novel theoretical perspective on the relationship between physical exercise and negative emotions, elucidating the associations among physical exercise, self-efficacy, smartphone addiction, and negative emotions. The findings suggest that physical exercise may regulate negative emotions by enhancing self-efficacy and mitigating smartphone addiction, thereby providing empirical support for mental health intervention practices within the university student population.

## Introduction

Negative emotion refers to the adverse effects resulting from life events, leading to emotional instability and subsequent disruptions to internal mental energy and endocrine balance [1]. Depression, anxiety, and pressure are prominent negative emotions experienced by college students [2]. Of these, anxiety is a frequently observed manifestation of psychopathology in adolescents, negatively impacting academic performance, social interactions, and adaptive functioning, and increasing the risk of mental health issues in adulthood [3]. Currently, the prevalence of negative emotions among college students in China is on the rise, and finding practical solutions is a pressing concern. Numerous studies indicate that the incidence of negative emotions increases significantly during adolescence, second only to behavioural disorders [4,5]. College students often experience substantial pressure related to academic performance and grades. Increasing academic workloads, uncertainty about future career paths, and concerns about the job market can contribute to anxiety, irritability, and excessive worry, exacerbating their anxiety and pressure. Prolonged exposure to such stress can easily result in chronic stress responses [2]. According to the Chinese College Student Mental Health Report, 54.72% of college students exhibit symptoms of anxiety, reflecting the widespread prevalence of negative emotions among contemporary college students and the significant risks they pose to mental health.

Physical exercise has garnered significant attention in recent years as a vital intervention strategy. Existing research demonstrates a significant negative correlation between physical exercise and negative emotions, indicating that increased physical activity is associated with a reduction in negative affect. [6]. Furthermore, studies have indicated that engaging in physical exercise can promote the release of endorphins in the brain, thereby improving mood and enhancing well-being [7]. Moreover, physical exercise has been shown to improve sleep quality and mitigate stress-related physiological responses, further reducing the likelihood of experiencing negative emotions. [8,9].However, while previous studies have established the beneficial effects of physical exercise on negative emotions, research exploring the underlying mechanisms remains limited. Existing studies primarily focus on the direct

relationship between physical exercise and negative emotions, with fewer investigations into the chain-mediating effects of how physical exercise indirectly influences negative emotions through psychological mediating variables, such as self-efficacy and smartphone addiction. This study aims to address this gap by delving into the intricate relationships between physical exercise, self-efficacy, smartphone addiction, and negative emotions, thereby providing a more comprehensive understanding of the positive impact of physical exercise on the psychological well-being of university students.

Physical exercise involves structured physical activity aimed at improving health, utilizing various methods and natural forces [10]. In recent years, declining physical health among college students, often due to unhealthy lifestyles, has contributed to reduced physical fitness and increased negative emotions [11,12]. College students are in a period of rapid physical development and emotional fluctuation [13], making their physical and mental health a key focus of academic research [14]. Through diverse training methods, physical exercise not only enhances physical fitness [15] but also stimulates the release of beneficial biochemical substances, such as endorphins, which help alleviate negative emotions [16]. Therefore, beyond its benefits for physical health, the impact of physical exercise on mental well-being is significant [17]. Research indicates that physical exercise can reduce negative emotions and lower the risk of mental disorders [18], while also mitigating anxiety-related psychophysiological responses, enhancing well-being, and preventing psychological problems, thereby reducing the likelihood of negative emotions occurring within the general population. [19]. To a certain extent, it mitigates the effects of negative emotions [20]. Further studies suggest that physical exercise can directly influence the frequency of smartphone use among college students [21] and has a moderating effect on their negative emotions [22].This study will focus on exploring the indirect influence of physical exercise on negative emotions, and, based on this, will construct a more comprehensive theoretical framework. Previous research has predominantly focused on the direct association between physical exercise and negative emotions. In contrast, this study will delve into the complex relationships among physical exercise, self-efficacy, smartphone addiction, and negative emotions, constructing a chain-mediating model. It will specifically investigate the independent mediating roles of self-efficacy and smartphone addiction, as well as their sequential mediating effects, with the aim of more comprehensively elucidating the underlying mechanisms through which physical exercise impacts the psychological well-being of university students. Based on these findings, Hypothesis H1 is proposed: Physical exercise has an adverse predictive effect on negative emotions.

Self-efficacy plays a crucial role in the daily lives of college students. As a core construct in Bandura's Social Cognitive Theory (SCT) [23], it refers to an individual's belief in their ability to organize and execute actions to achieve specific goals. This self-assessment significantly influences activity selection and effort investment [24]. Self-efficacy can determine individuals' choices of activities and their persistence in them, shaping their attitudes towards challenges, the acquisition of new behaviours, and emotional responses during tasks [23]. Different stages of behavioural engagement require distinct types of self-efficacy – for instance, when adopting exercise habits versus overcoming obstacles to maintaining them [25]. Self-efficacy is a crucial variable for achieving successful performance and is essential for completing tasks effectively. Moreover, an individual's self-efficacy is negatively correlated with negative emotions and positively correlated with successful experiences [26]. The primary function of self-efficacy is behavioural regulation, reflected in the strength of one's self-confidence. Individuals with high self-efficacy tend to exert greater perseverance and effort in achieving their goals. In contrast, those with low self-efficacy are more prone to self-doubt, questioning their capabilities in the face of failure, and often settling for mediocrity [27]. A substantial body of research has demonstrated that physical exercise significantly enhances individuals' self-efficacy. Engaging in aerobic exercise can bolster individuals' sense of bodily control and confidence, consequently elevating self-efficacy [28]. Furthermore, high levels of self-efficacy have been shown to be an effective protective factor against negative emotions, as it enables individuals to better manage stress and challenges, thereby reducing the risk of anxiety and depression [29].Currently, enhancing self-efficacy through physical exercise has become a vital strategy for personal health development, offering a potential solution to low self-efficacy [30]. Surveys indicate that only 50.4% of college students exhibit high self-efficacy after returning from holidays [31]. Given that research now links positive self-efficacy to both personal competence and mental health [32], further investigation into

the relationship between self-efficacy and physical exercise is warranted. This study posits that physical exercise has a positive impact on self-efficacy, which in turn can alleviate negative emotions. Consequently, we constructed a mediation model to examine the mediating role of self-efficacy in the relationship between physical exercise and negative emotions. This relationship is theoretically sound, as physical exercise can enhance individuals' confidence in their abilities, thereby improving their capacity to cope with negative feelings. Therefore, this study proposes Hypothesis H2: Self-efficacy mediates the relationship between physical exercise and negative emotions.

The Impact of Smartphone Addiction on College Students' Negative Emotions in China and the Urgent Need for Solutions. Smartphone addiction, also referred to as mobile phone dependence, is characterized by the excessive use of smartphones and a loss of behavioural control, leading to significant cognitive preoccupation, emotional changes, and relapse [33]. While smartphones offer convenience [34], they can negatively impact physical and mental health, contributing to obesity, insomnia, depression, and anxiety [35,36]. Similar to gambling and video game addiction, smartphone addiction exhibits strong psychological and behavioural effects. Individuals struggling with uncontrolled smartphone addiction often report reduced life satisfaction and heightened negative emotions [37]. According to the 51st Statistical Report on Internet Development in China released by the China Internet Network Information Centre [38], China had 1.051 billion internet users as of December 2022, of which 1.047 billion were mobile phone users, representing 99.6% of the total user base. Data analysis indicates that smartphone addiction is highly prevalent among college students, with detection rates ranging from 21.4–27.4% [39]. In 2022, the smartphone addiction rate among Chinese college students was approximately 36.6% [40], highlighting smartphone addiction as the most direct negative consequence of excessive smartphone use [41]. Therefore, effectively curbing the rising rate of smartphone addiction has become an urgent public health concern [42].Excessive smartphone use not only consumes considerable time, leading to reduced engagement in academic and social activities, but also precipitates feelings of loneliness, anxiety, and depression. Research indicates that individuals addicted to their mobile phones are more prone to experiencing sleep disturbances, which, in turn, exacerbate negative emotions [43,44]. Moreover, smartphone addiction may also disrupt emotional regulation by affecting dopamine pathways, thereby rendering individuals more susceptible to the intrusion of negative emotions [45]. Extensive research suggests a close relationship between physical exercise and smartphone addiction [46]. Moreover, physical exercise can alleviate negative emotions, such as anxiety, depression, and distress, linked to smartphone addiction, thereby promoting mental well-being [47]. However, a consensus on the mechanisms through which physical exercise influences smartphone addiction remains elusive within the academic community. Therefore, this study not only focuses on the direct influence of physical exercise on negative emotions but also emphasises exploring the mediating role of smartphone addiction. It will also consider the specific characteristics of university students within the Chinese cultural context, such as the impact of academic pressure and interpersonal relationships on the study's findings. Furthermore, it will integrate both Social Cognitive Theory and Self-Determination Theory, to examine the effects of physical exercise on negative emotions from multiple theoretical perspectives, with the aim of gaining a deeper understanding of the overall impact of physical exercise on the psychological well-being of university students. Theoretically, physical exercise can serve as a substitute behaviour, reducing reliance on smartphones. By improving physical and mental well-being, physical exercise may diminish the need for individuals to use their phones to cope with negative emotions, consequently reducing smartphone addiction. A decrease in smartphone addiction will, in turn, further alleviate negative emotions. Therefore, this study hypothesizes that smartphone addiction mediates the relationship between physical exercise and negative emotions. This study will explore this mediation effect and analyse the specific characteristics of Chinese university students within this process. Thus, this study proposes Hypothesis H3: Smartphone addiction mediates the relationship between physical exercise and negative emotions.

In contrast to Western countries, traditional Chinese culture emphasises collectivism, harmonious interpersonal relationships, and the importance of academic achievement [48]. These cultural factors may exert an influence on university students' participation in physical exercise, the development of self-efficacy, the use of smartphones, and the experience

of negative emotions. Students from a collectivist cultural background may be more inclined to engage in group-based physical activities, which could shape their perceptions and attitudes towards exercise. An overemphasis on academic performance may also give rise to academic pressure, subsequently affecting students' negative emotions and smartphone usage.

This study selected Henan and Jiangsu provinces as the research sample because these provinces are representative of the national economic development and educational resources. Jiangsu Province is economically developed with abundant educational resources, while Henan Province is a populous province with rapidly developing higher education. The university student populations in these two provinces are representative in terms of social background, lifestyle, and academic pressure, allowing for a reflection of the general circumstances of Chinese university students. Furthermore, the two provinces exhibit certain differences in geographical location and climatic conditions, which may influence the type and frequency of physical exercise undertaken by university students, thereby providing a more diverse perspective for the research. For instance, the higher level of urbanisation in Jiangsu Province may lead to a greater preference for indoor physical activities among university students, whereas outdoor physical activities may be more prevalent in Henan Province, which has a larger, less densely populated area. Finally, by selecting these two provinces, the study aims to effectively control research costs and ensure the quality and comparability of the data.

Grounded in Bandura's SCT [23], this study explores the impact of physical exercise on negative emotions. SCT posits that behaviour is shaped by the reciprocal interplay of personal factors (e.g., cognition, self-efficacy), environmental factors, and behaviour [23]. Within this framework, self-efficacy is a core concept, defined as an individual's belief in their capability to execute specific behaviours successfully [23]. Individuals with high self-efficacy typically demonstrate stronger self-control, which enhances their ability to cope with negative emotions and thereby reduces the likelihood of maladaptive behaviours [49]. In this study, physical exercise has been shown to improve self-efficacy significantly [50] and mitigate negative emotions associated with smartphone addiction [18]. Simultaneously, enhanced self-efficacy may indirectly reduce the experience of negative emotions. Smartphone addiction not only exacerbates negative emotions but also can impair an individual's self-control [51], whereas physical exercise can alleviate these adverse effects by promoting positive emotional states [42]. SCT offers a robust theoretical foundation for examining the mechanisms by which physical exercise, smartphone addiction, and self-efficacy influence negative emotions. The SCT highlights the dynamic interaction between behaviour, emotions, and cognition, thus providing a theoretical basis for the chained mediation model proposed in this study. According to SCT, individual behaviours, such as excessive mobile phone use, are also influenced by self-efficacy. If individuals lack self-efficacy in coping with stress, they may be more likely to rely on their phones as a means of escape or for managing negative emotions, potentially leading to addiction. Therefore, this study will examine how physical exercise indirectly affects university students' negative emotions by enhancing self-efficacy and reducing smartphone addiction. Furthermore, this study introduces Self-Determination Theory (SDT) as a supplementary theoretical framework to more comprehensively explain how physical exercise promotes psychological well-being by satisfying individuals' basic psychological needs – autonomy, competence, and relatedness [52]. SDT posits that physical exercise can strengthen individuals' intrinsic motivation and self-determination, thereby decreasing dependence on external stimuli (such as smartphones) and improving emotional regulation capabilities [52].Drawing upon SCT and SDT, this study posits that physical exercise alleviates negative emotions by enhancing self-efficacy and reducing smartphone addiction. According to SCT, physical exercise can boost individuals' self-efficacy, empowering them with greater ability to cope with stress and decrease their reliance on smartphones. SDT explains how physical exercise satisfies individuals' psychological needs and strengthens intrinsic motivation, thereby reducing smartphone dependence. Therefore, self-efficacy and smartphone addiction are likely to form a chain mediation effect, collectively influencing negative emotions. By integrating SCT and SDT, this study constructs a chain-mediating model, aiming to elucidate the psychological mechanisms by which physical exercise alleviates negative emotions, acting through enhanced self-efficacy (SCT) and reduced smartphone addiction (SDT). Through the integration of SCT and SDT, this study not only deepens the understanding

of the mechanisms between physical exercise and negative emotions but also provides a richer theoretical foundation for subsequent intervention practices. Especially in the context of the proliferation of smartphones and increasing academic pressure, adopting a dual perspective of motivation and self-regulation helps to design more effective strategies for promoting mental health. Accordingly, Hypothesis H4 is proposed: Self-efficacy and smartphone addiction sequentially mediate the relationship between physical exercise and negative emotions (Fig 1).

## Methods

### Data sources

The study received ethical approval from the Ethics Committee of Nantong University, China. Informed consent was obtained from all participants involved in the study. The data for this study were obtained from the 2022 cross-sectional survey of the "CPAHLS-CS". Data collection adhered to standardised procedures to ensure reproducibility. A multi-stage stratified cluster sampling method was employed. Based on the representativeness of economic development and educational resources, Jiangsu Province in East China and Henan Province in Central China were selected in the first stage. In the second stage, one undergraduate institution and one vocational college were randomly selected from both the provincial capital and several prefecture-level cities in each province, resulting in a total of eight universities. In the third stage, classes were randomly selected for participation within each selected university, stratified by year and major.

Informed consent was obtained from all the subjects involved in the study. The study questionnaire followed the American Psychological Association (APA) Ethical Principles of Psychologists and Code of Conduct (APA Ethical Principles of Psychologists and Code of Conduct). The study was completed anonymously. The collection of general demographic information for this study included school, gender, and grade level. Before the researcher administered the questionnaire, the researcher read the beginning of the questionnaire and informed the participant of the relevant informed consent and completion requirements, and after the participant verbally agreed, the researcher administered the questionnaire. After the participant read the beginning of the questionnaire again, if the participant was willing to continue to fill in. The study protocol for this study received approval from the ethics committee at Nantong University and was documented under approval number 2022(70).

This study focuses on data from universities in Henan Province (Central China) and Jiangsu Province (East China), both of which are significant development regions in China, with high population quality and education levels, thus being highly representative. Specifically, one undergraduate and one vocational institution were selected from the provincial

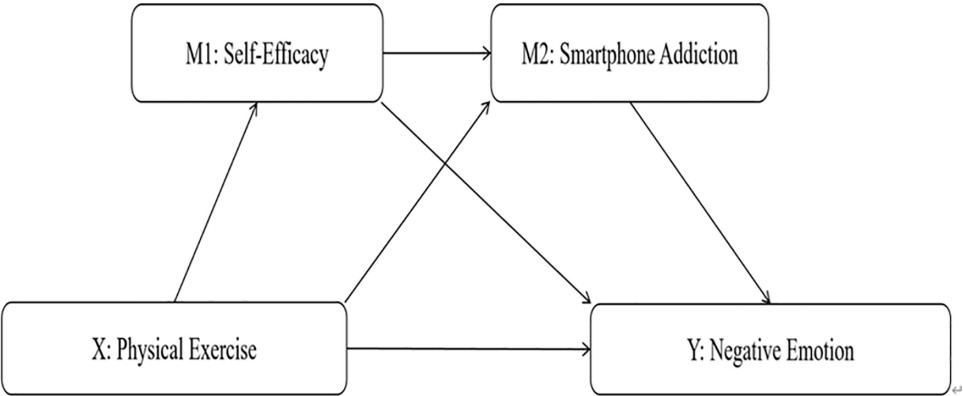

**Fig 1. Schematic diagram of the chain mediation model between physical exercise, self-efficacy, smartphone addiction and negative emotions.** Note: Path coefficients indicate that physical exercise indirectly reduces negative emotions by enhancing self-efficacy and decreasing smartphone addiction.

capital and prefecture-level cities within each province, resulting in a total of four universities per province, and eight universities overall. Following the removal of cases with missing core variables, the final sample comprised 16,355 valid cases. The recruitment period for the study spanned from 08/10/2024 to 09/11/2024, with the subsequent questionnaire survey being conducted from 11/11/2024 to 24/11/2024. The distribution of the survey participants is presented in **Table 1**.

While the sample exhibited some imbalance in the distribution of gender and grade level, we mitigated potential sampling biases through stratified analysis and weighting procedures. Specifically, in subsequent statistical analyses, data were weighted to account for the distribution of gender and grade level, ensuring the representativeness of each subgroup. Furthermore, demographic variables were controlled for in regression analyses and mediation effect tests. These measures enhanced the reliability and generalizability of our findings.

### Variable measurement

**Physical exercise.** The PARS-3, initially developed by Kimio Hashimoto and later revised by Liang et al., is primarily employed to assess the engagement of college students in physical exercise. It evaluates exercise volume across three dimensions: the intensity, frequency, and duration of each session, thus serving as an indicator of physical exercise participation levels [53]. When completing the questionnaire, respondents rate three items—intensity, frequency, and duration—each on a 5-point scale, ranging from "1" (never engaging in physical exercise) to "5". Higher scores reflect a greater volume of exercise, representing the physical exercise behaviour of college students over a specific period. According to the Chinese adult norm grading standards for the PARS-3, exercise levels are defined as follows: low exercise volume (≤19 points), moderate exercise volume (20–42 points), and high exercise volume (≥43 points) [53]. The raw score is calculated using the formula: Exercise volume score = intensity × (time – 1) × frequency. A higher total score indicates both greater exercise intensity and a higher level of participation. In this study, the scale demonstrated a Cronbach's α coefficient of 0.72, indicating acceptable internal consistency. Previous research reported a test-retest reliability of 0.82 for this scale [54].To further validate the psychometric properties of the scale within the current sample, confirmatory factor analysis (CFA) was performed. The results indicated a good fit for the three-factor model: $\chi^2/df = 2.85$, CFI = 0.95, TLI = 0.93, RMSEA = 0.057 (90% CI: 0.054–0.060), SRMR = 0.041, thereby supporting the structural validity of the scale. Furthermore, all item factor loadings ranged from 0.65 to 0.82, exceeding the commonly accepted threshold of 0.6, which further demonstrates the scale's robust structural validity.

**Negative emotion.** Negative emotion refers to unpleasant, dissatisfying, or uncomfortable feelings, encompassing sadness, anger, fear, shame, jealousy, guilt, anxiety, and disgust. As a significant component of the emotional spectrum,

**Table 1. Distribution of demographic characteristics in the sample population (N = 16,355).**

| Variable | | n | % |
|---|---|---|---|
| Gender | Male | 6474 | 39.5 |
| | Female | 9906 | 60.5 |
| Grade | Freshmen | 5565 | 33.9 |
| | Sophomores | 7586 | 46.3 |
| | Juniors | 2282 | 13.9 |
| | Seniors | 947 | 5.9 |
| Region | East China (Jiangsu Province) | 10184 | 62.2 |
| | Central China (Henan Province) | 6755 | 37.8 |
| Total | | 16355 | 100 |

The sample comprised a higher proportion of female respondents (60.5%), with Sophomores accounting for the largest share (46.3%). A greater number of respondents hailed from the East China region (62.2%).

it profoundly influences an individual's mental health and behavioural choices. The DASS is based on Clark and Watson's tripartite model, which posits that depression, anxiety, and pressure share both distinct and overlapping symptomatic features [55], making it suitable for collectively assessing negative emotion [56]. In this study, the simplified Chinese version of the DASS-C21, initially developed by Lovibond et al., revised by Antony et al., and translated by Wen, was utilized. The scale comprises three subscales: depression, anxiety, and pressure, each containing seven items. Responses are scored on a 4-point Likert scale (0–3), with higher scores indicating more severe negative emotion. The DASS-C21 demonstrates high reliability and validity among Chinese college students. The scale's Cronbach's α coefficient and test-retest reliability were 0.912 and 0.751, respectively, with an average inter-item correlation of 0.338. The Pearson correlation coefficients between the three subscale scores and the total DASS-C21 score ranged from 0.895 to 0.910, while correlations between subscales ranged from 0.708 to 0.741 ($p < 0.01$). Confirmatory factor analysis indicated good model fit, with the following indices: GFI = 0.91, AGFI = 0.89, CFI = 0.91, IFI = 0.91, TLI = 0.89, and RMSEA = 0.06 [57].In this study sample, the Cronbach's α coefficient for the total DASS-C21 scale was 0.917. The α coefficients for each dimension were as follows: Stress 0.854, Anxiety 0.843, and Depression 0.861, indicating good internal consistency of the scale. Confirmatory factor analysis (CFA) showed good model fit: $\chi^2$/df = 3.82, CFI = 0.93, TLI = 0.91, RMSEA = 0.064 (90% CI: 0.061–0.067), SRMR = 0.045, further supporting the structural validity of the scale.

**Self-efficacy.**  In this study, college students' self-efficacy in physical exercise was assessed using the PASS, developed by Jiang et al. [58]. The PASS comprises two dimensions: situational motivation and subjective support, encompassing a total of 10 items. Responses were recorded on a 5-point Likert scale, ranging from "1 = strongly disagree" to "5 = strongly agree." Higher scores indicate greater self-efficacy in physical exercise. The cumulative variance contribution rate of the two factors was 67.782%, and confirmatory factor analysis confirmed a good fit for the two-factor model. The total scale demonstrated a Cronbach's α coefficient of 0.908, a split-half reliability of 0.819, and a test-retest reliability of 0.531 [58]. In this study sample, the Cronbach's α coefficient for the total PASS scale was 0.901. The α coefficients for the two dimensions of situational motivation and subjective support were 0.876 and 0.832, respectively. Confirmatory factor analysis (CFA) results showed good model fit: $\chi^2$/df = 4.15, CFI = 0.94, TLI = 0.92, RMSEA = 0.068 (90% CI: 0.065–0.071), SRMR = 0.049, indicating that the scale had good structural validity in this study.

**Smartphone addiction.**  The MPATS was employed to measure the level of smartphone addiction among college students. The Smartphone Addiction Tendency Scale, developed by Chinese scholars Xiong et al. [59], was used. The scale consists of four dimensions: withdrawal symptoms (adverse physiological or psychological reactions when not engaged in smartphone activities), salient behaviour (the dominance of smartphone use in thoughts and behaviours), social comfort (the role of smartphones in interpersonal communication), and mood alteration (emotional changes caused by smartphone use), with a total of 16 items. Each item is scored using a five-point Likert scale, ranging from "completely disagree" to "completely agree," with corresponding scores of 1–5. The maximum total score is 80, while the minimum is 16. Higher scores indicate a greater tendency towards smartphone addiction, while lower scores suggest a weaker tendency. The factor loadings for the four dimensions ranged from 0.51 to 0.79, with a variance explanation rate of 54.3%. Confirmatory factor analysis results supported the suitability of the four-factor model. The Cronbach's alpha coefficient for the full scale was 0.83, with coefficients for the four factors ranging from 0.55 to 0.80. The test-retest reliability of the full scale was 0.91, while the values for the four factors ranged from 0.75 to 0.85 [59]. In this study sample, the Cronbach's α coefficient for the total MPATS scale was 0.891. The α coefficients for the four dimensions were as follows: Withdrawal symptoms 0.802, Salience behaviour 0.785, Social comfort 0.763, and Mood alteration 0.779. Confirmatory factor analysis (CFA) results showed good fit for the four-factor model: $\chi^2$/df = 4.62, CFI = 0.92, TLI = 0.90, RMSEA = 0.071 (90% CI: 0.068–0.074), SRMR = 0.051, supporting the structural validity of the scale (Table 2).

In this study, the severity of pressure, depression, and anxiety symptoms was categorized into three levels: "no risk," "moderate risk," and "high risk." Specifically, the "normal" classification in the DASS-21 was designated as "no risk," while

**Table 2. College students' physical exercise level and negative emotional risk assessment list (n = 16355).**

| Category | | Total | Physical Exercise: Low | Middle | High | Statistic | Pressure: Risk-free | Medium risk | High risk | Statistic | Anxiety: Risk-free | Medium risk | High risk | Statistic | Depression: Risk-free | Medium risk | High risk | Statistic |
|---|---|---|---|---|---|---|---|---|---|---|---|---|---|---|---|---|---|---|
| **Generalize** | | | | | | | | | | | | | | | | | | |
| Total | n | 16355 | 12547 | 2236 | 1597 | | 8421 | 7211 | 723 | | | 10481 | 5874 | | 3516 | 10806 | 2033 | |
| Total | % | 100.00% | 76.70% | 13.70% | 9.60% | | 51.40% | 44.10% | 4.50% | | | 64.10% | 35.90% | | 21.40% | 66.10% | 12.50% | |
| **Region** | | | | | | | | | | | | | | | | | | |
| East China | n | | 7440 | 1398 | 940 | 53.26 | 5048 | 4248 | 482 | 106.51 | | 6196 | 3582 | 96.96 | 2267 | 6263 | 1248 | 102.43 |
| East China | % | | 59.30% | 62.50% | 58.90% | | 59.90% | 58.80% | 66.70% | | | 59.10% | 61.00% | | 64.50% | 58.00% | 61.40% | |
| Central China | n | | 5107 | 838 | 657 | | 3383 | 2978 | 241 | | | 4285 | 2292 | | 1249 | 4543 | 785 | |
| Central China | % | | 40.70% | 37.50% | 41.10% | | 40.10% | 41.20% | 33.30% | | | 40.90% | 39.00% | | 35.50% | 42.00% | 38.60% | |
| **Gender** | | | | | | | | | | | | | | | | | | |
| Male | n | | 3886 | 1309 | 1279 | 2056.66 | 3083 | 2886 | 505 | 526.02 | | 3803 | 2658 | 517.67 | 1401 | 3880 | 1180 | 522.44 |
| Male | % | | 31.00% | 58.50% | 80.10% | | 36.60% | 39.90% | 69.80% | | | 36.30% | 45.30% | | 39.80% | 35.90% | 58.00% | |
| Female | n | | 8661 | 927 | 318 | | 5348 | 4340 | 218 | | | 6678 | 3216 | | 2115 | 6926 | 853 | |
| Female | % | | 69.00% | 41.50% | 19.90% | | 63.40% | 60.10% | 30.20% | | | 63.70% | 54.70% | | 60.20% | 64.10% | 42.00% | |
| **Grade** | | | | | | | | | | | | | | | | | | |
| Freshman | n | | 4268 | 823 | 474 | 214.32 | | | | 353.75 | | 3915 | 1650 | 373.44 | 1349 | 3778 | 438 | 414.33 |
| Freshman | % | | 34.00% | 36.80% | 29.70% | | | | | | | 37.40% | 28.10% | | 38.40% | 35.00% | 21.50% | |
| Sophomores | n | | 5860 | 1002 | 724 | | | | | | | 4558 | 3026 | | 1536 | 4930 | 1118 | |
| Sophomores | % | | 46.70% | 44.80% | 45.30% | | | | | | | 43.50% | 51.50% | | 43.70% | 45.60% | 55.00% | |
| Juniors | n | | 1690 | 288 | 304 | | | | | | | 1462 | 817 | | 483 | 1483 | 313 | |
| Juniors | % | | 13.50% | 12.90% | 19.00% | | | | | | | 13.90% | 13.90% | | 13.70% | 13.70% | 15.40% | |
| Seniors | n | | 729 | 123 | 95 | | | | | | | 546 | 381 | | 148 | 615 | 164 | |
| Seniors | % | | 5.80% | 5.50% | 5.90% | | | | | | | 5.20% | 6.50% | | 4.20% | 5.70% | 8.10% | |
| **Age** | | | | | | | | | | | | | | | | | | |
| 17 | n | | 585 | 87 | 54 | 253.01 | 409 | 309 | 8 | 210.14 | | 503 | 223 | 301.82 | 152 | 527 | 47 | 399.76 |
| 17 | % | | 4.70% | 3.90% | 3.40% | | 4.90% | 4.30% | 1.10% | | | 4.80% | 3.80% | | 4.30% | 4.90% | 2.30% | |
| 18 | n | | 3187 | 624 | 349 | | 2361 | 1681 | 118 | | | 2889 | 1271 | | 1013 | 2806 | 341 | |
| 18 | % | | 25.40% | 27.90% | 21.90% | | 28.00% | 23.30% | 16.30% | | | 27.60% | 21.60% | | 28.80% | 26.00% | 16.80% | |
| 19 | n | | 4137 | 661 | 490 | | 2747 | 2312 | 229 | | | 3383 | 1905 | | 1145 | 3509 | 634 | |
| 19 | % | | 33.00% | 29.60% | 30.70% | | 32.60% | 32.10% | 31.70% | | | 32.30% | 32.40% | | 32.60% | 32.50% | 31.20% | |
| 20 | n | | 2914 | 533 | 406 | | 1848 | 1780 | 225 | | | 2321 | 1532 | | 778 | 2460 | 615 | |
| 20 | % | | 23.20% | 23.80% | 25.40% | | 21.90% | 24.70% | 31.10% | | | 22.10% | 26.10% | | 22.10% | 22.80% | 30.30% | |
| 21 | n | | 1230 | 218 | 214 | | 751 | 806 | 105 | | | 993 | 669 | | 307 | 1085 | 270 | |
| 21 | % | | 9.80% | 9.70% | 13.40% | | 8.90% | 11.20% | 14.50% | | | 9.50% | 11.40% | | 8.70% | 10.00% | 13.30% | |
| 22 | n | | 480 | 106 | 80 | | 305 | 323 | 38 | | | 392 | 274 | | 121 | 419 | 126 | |
| 22 | % | | 3.80% | 4.70% | 5.00% | | 3.60% | 4.50% | 5.30% | | | 3.70% | 4.70% | | 3.40% | 3.90% | 6.20% | |

Scores within the "normal" range on the DASS-C21 scale were classified as "risk-free" in this study; scores within the "mild" and "moderate" ranges were categorised as "medium risk"; and scores indicating a "severe" state on the DASS-C21 scale were designated as "high risk" in this study. Moreover, students in East China exhibit a higher proportion of high-risk negative emotions; female students demonstrate higher rates of high-risk stress, anxiety, and depression than their male counterparts; and Seniors show significantly higher risks of smartphone addiction and depression compared to their junior peers.

the "mild" and "moderate" classifications were grouped as "moderate risk." The "severe" and "extremely severe" classifications were combined into a single "high risk" category.

## Statistical analysis

In this study, data processing and analysis were primarily conducted using SPSS 27.0 and Excel software, adhering to rigorous procedures to ensure the reliability and validity of the findings. Prior to the mediation analyses, descriptive statistics and correlation analyses were performed on all variables to examine their basic relationships. Specifically, Pearson correlation coefficients were calculated to assess the linear relationships between physical exercise, self-efficacy, smartphone addiction, and negative emotions. 1) Initially, Excel was employed for the preliminary processing of data collected via the Questionnaire Star platform, including the identification and removal of incomplete or anomalous responses. 2) Subsequently, common method bias was assessed using SPSS, specifically through Harman's single-factor test, which performed exploratory factor analysis on all questionnaire items related to physical exercise, negative emotion, self-efficacy, and smartphone addiction. The results revealed that the largest factor accounted for only 35.611% of the variance among the extracted principal components, which fell below the 40% threshold, indicating that there was no significant common method bias in this study. 3) Following this, an in-depth analysis of the collected data from college students was conducted. Chi-square tests were utilized to examine differences in negative emotion across age, grade, gender, and region, with effect sizes interpreted using Cohen's d criteria ($\eta^2$ ranging from 0 to 1, where 0.01, 0.06, and 0.14 represent small, medium, and large effects, respectively). Specifically, $0.2 \leq |\eta^2| < 0.5$ indicates a small effect, $0.5 \leq |\eta^2| < 0.8$ a medium effect, and $|\eta^2| \geq 0.8$ a large effect. Cramer's V, ranging from 0 to 1, reflects the strength of association between variables, with higher values denoting stronger associations: $0.0 \leq V < 0.1$ suggests no or very weak association, $0.1 \leq V < 0.3$ weak association, $0.3 \leq V < 0.5$ moderate association, and $V \geq 0.5$ strong association [60]. 4) Pearson correlation analysis was then employed to explore the relationships among physical exercise, negative emotion, self-efficacy, and smartphone addiction. Prior to conducting multiple linear regression analyses, we assessed the presence of multicollinearity among the independent variables by calculating the Variance Inflation Factor (VIF) and tolerance values. A VIF value exceeding 10, or a tolerance value below 0.1, indicated a severe multicollinearity issue. To address potential multicollinearity, we employed a strategy of removing highly correlated variables. To minimise potential biases inherent in self-reported data, specifically recall and social desirability biases, several strategies were implemented throughout the questionnaire design and data collection process. All questionnaires utilised validated, established instruments, deemed appropriate for use within the Chinese undergraduate population. Questionnaire completion was anonymised, with clear instructions emphasising the exclusive use of data for scientific purposes and the guarantee of absolute confidentiality, thereby reducing the potential for social desirability effects. Item phrasing was designed to be unambiguous and non-leading, thus minimising comprehension bias. Data collection was conducted within a consolidated timeframe to preclude recall error arising from temporal intervals. Prior to statistical analysis, the dataset underwent rigorous cleaning and outlier treatment, which involved the exclusion of responses deemed implausible (e.g., identical selections across all items, patterned responding). 5) Mediation analyses were performed using the PROCESS macro (Model 6) developed by Hayes (2018) in SPSS. The model was constructed with physical exercise as the independent variable, negative emotions as the dependent variable, and self-efficacy and smartphone addiction as mediating variables, forming a parallel multiple mediation model. Model selection was based on theoretical foundations and prior research, hypothesizing that self-efficacy and smartphone addiction could function as either independent mediators or a serial (chain) mediator influencing the relationship between physical exercise and negative emotions. The significance of the mediation effects was estimated using the bootstrapping method (with 5000 resamples). A significant mediation effect was determined if the 95% confidence intervals (CIs), based on these bootstrapped estimates, did not include zero.

 

### Inclusivity in global research

Additional information regarding the ethical, cultural, and scientific considerations specific to inclusivity in global research is included in the Supporting Information.

## Results

### Descriptive results analysis

As shown in **Table 3**, significant differences were observed across all four variables at the gender level ($p < 0.001$), with the most pronounced distinction between male and female college students in physical exercise ($\eta^2 = 0.112$). Specifically, males exhibited a significantly higher mean score on the "situational motivation" dimension of physical exercise (M = 22.69) compared to females (M = 9.59), and also demonstrated a higher proportion of individuals experiencing high-risk stress. At the grade level, all four variables exhibited significant differences ($p < 0.001$), with the most considerable disparity among freshmen, sophomores, juniors, and seniors observed in the salient behaviour sub-dimension of smartphone addiction ($\eta^2 = 0.034$). Seniors exhibited a significantly higher mean score on the "salience" dimension of smartphone addiction (M = 17.14) compared to Freshman (M = 15.75), and also demonstrated a greater proportion of individuals at high risk of depression. Regarding age, significant differences were absent only in the self-efficacy sub-dimensions ($p = 0.01$) among the four variables. University students aged 17 demonstrated generally lower levels of negative emotions compared to students in other age groups, whilst exhibiting no significant difference in self-efficacy. At the regional level, significant differences were found in the investment of effort in physical exercise ($p = 0.044$). In contrast, no significant differences were detected in the three sub-dimensions of negative emotion—pressure, depression, and anxiety, with all $\eta^2$ values below 0.002.

### Correlation analysis

Based on the data presented in **Table 4**, physical exercise and its sub-dimensions demonstrated significant positive correlations with self-efficacy and its sub-dimensions ($r = 0.298$–$0.303$). Furthermore, bodily exercise was negatively correlated with both smartphone addiction ($r = -0.124$ to $-0.065$) and negative emotion ($r = -0.057$ to $-0.046$). In addition, negative emotion exhibited positive correlations with both self-efficacy ($r = 0.001$–$0.045$) and smartphone addiction ($r = 0.464$–$0.917$). Conversely, self-efficacy displayed a significant negative correlation with smartphone addiction ($r = -0.122$ to $-0.023$).

### Regression analysis

The results are presented in **Table 5**. After controlling for gender, grade, region, and age, physical exercise was found to negatively predict smartphone addiction ($\beta = -0.092$, $p < 0.001$), pressure ($\beta = -0.156$, $p < 0.001$), anxiety ($\beta = -0.02$, $p < 0.001$), and depression ($\beta = -0.196$, P < 0.001). Self-efficacy positively predicted pressure ($\beta = 0.041$, $p < 0.001$), anxiety ($\beta = 0.050$, $p < 0.001$), and depression ($\beta = 0.038$, $p < 0.001$). Smartphone addiction positively predicted pressure ($\beta = 0.185$, $p < 0.001$), anxiety ($\beta = 0.179$, $p < 0.001$), and depression ($\beta = 0.182$, $p < 0.001$).

### Mediation effect test

As shown in **Table 6**, the 95% CI for smartphone addiction in the relationship between physical exercise and pressure was [−0.019, −0.014]. This confirms that smartphone addiction mediated the relationship between physical exercise and pressure in this study. Similarly, smartphone addiction also mediated the relationship between physical exercise and anxiety, as indicated by a 95% CI of [−0.019, −0.014]. The mediating role of smartphone addiction was further confirmed in the relationship between physical exercise and depression, with a 95% CI of [−0.019, −0.014]. In addition, self-efficacy mediated the relationship between physical exercise and pressure, as evidenced by a 95% CI of [0.004, 0.006]. The mediating effect of self-efficacy was also supported in the relationships between physical exercise and anxiety (95% CI: [0.005, 0.007]) and between physical exercise and depression (95% CI: [0.004, 0.006]). Furthermore, the chain mediation

**Table 3. Descriptive statistics and intergroup differences analysis for each variable across different demographic characteristics.**

| | | Physical Exercise | Self-Efficacy | | | Negative Emotion | | | Smartphone Addiction | | | | |
|---|---|---|---|---|---|---|---|---|---|---|---|---|---|
| | | | Situational motivation | Subjective support | Total score | Pressure | Anxiety | Depression | Withdrawal symptoms | Highlight behaviour | Social comfort | Mood changes | Total score |
| **Questionnaire Filling Areas** | | | | | | | | | | | | | |
| East China | M | 14.84 | 22.371 | 15.56 | 37.92 | 12.8 | 12.42 | 12.101 | 16.57 | 9.01 | 8.01 | 7.31 | 40.9 |
| | sd | 18.941 | 5.501 | 3.579 | 8.802 | 5.29 | 5.14 | 5.283 | 6.008 | 4.219 | 3.317 | 3.206 | 15.426 |
| Central China | M | 14.6 | 21.521 | 15.2 | 36.72 | 12.79 | 12.27 | 12.101 | 16.45 | 8.93 | 7.93 | 7.13 | 40.43 |
| | sd | 19.42 | 5.101 | 3.33 | 8.092 | 4.932 | 4.771 | 4.948 | 5.642 | 3.978 | 3.187 | 3.06 | 14.447 |
| Total | M | 14.75 | 22.03 | 15.42 | 37.45 | 12.8 | 12.36 | 12.101 | 16.52 | 8.98 | 7.98 | 7.24 | 40.72 |
| | sd | 19.132 | 5.362 | 3.486 | 8.548 | 5.151 | 4.997 | 5.153 | 5.865 | 4.125 | 3.267 | 3.15 | 15.047 |
| | η² | 0 | 0.006 | 0.003 | 0.005 | 0 | 0 | 0 | 0 | 0 | 0 | 0.001 | 0 |
| | F | 0.585 | 94.202 | 39.496 | 74.829 | 0.008 | 3.19 | 4.734 | 1.465 | 1.607 | 2.553 | 13.147 | 3.706 |
| | P | 0.444 | <0.001 | <0.001 | <0.001 | 0.927 | 0.074 | 0.988 | 0.226 | 0.205 | 0.11 | <0.001 | 0.054 |
| **Gender** | | | | | | | | | | | | | |
| Male | M | 22.69 | 23.36 | 16.26 | 39.621 | 13.53 | 13.06 | 12.991 | 16.35 | 9.36 | 7.87 | 7.29 | 40.87 |
| | sd | 24.056 | 5.23 | 3.335 | 8.282 | 5.924 | 5.819 | 5.946 | 6.373 | 4.495 | 3.454 | 3.381 | 16.63 |
| Female | M | 9.59 | 21.17 | 14.87 | 36.041 | 12.33 | 11.91 | 11.531 | 16.63 | 8.73 | 8.05 | 7.21 | 40.62 |
| | sd | 12.675 | 5.27 | 3.473 | 8.423 | 4.519 | 4.324 | 4.474 | 5.509 | 3.846 | 3.137 | 2.991 | 13.925 |
| Total | M | 14.75 | 22.03 | 15.42 | 37.451 | 12.8 | 12.36 | 12.101 | 16.52 | 8.98 | 7.98 | 7.24 | 40.72 |
| | sd | 19.132 | 5.362 | 3.486 | 8.548 | 5.151 | 4.997 | 5.153 | 5.865 | 4.125 | 3.267 | 3.15 | 15.047 |
| | η² | 0.112 | 0.04 | 0.038 | 0.042 | 0.013 | 0.013 | 0.019 | 0.001 | 0.006 | 0.001 | 0 | 0 |
| | F | 1975.962 | 649.165 | 618.927 | 684.676 | 205.812 | 202.068 | 303.896 | 8.711 | 89.443 | 11.813 | 2.638 | 1.06 |
| | P | 0.003 | <0.001 | <0.001 | <0.001 | <0.001 | <0.001 | <0.001 | 0.003 | <0.001 | 0.001 | 0.104 | 0.303 |
| **Age** | | | | | | | | | | | | | |
| 17 | M | 12.94 | 21.52 | 15.18 | 36.701 | 12.08 | 11.68 | 11.33 | 16.11 | 7.97 | 8.01 | 7.15 | 39.25 |
| | sd | 16.324 | 5.302 | 3.528 | 8.494 | 4.173 | 3.956 | 4.156 | 5.47 | 3.711 | 3.183 | 3.017 | 13.483 |
| 18 | M | 14.36 | 22.09 | 15.58 | 37.671 | 12.2 | 11.8 | 11.32 | 16.13 | 8.22 | 7.74 | 6.91 | 39 |
| | sd | 17.891 | 5.288 | 3.405 | 8.369 | 4.667 | 4.447 | 4.583 | 5.709 | 3.821 | 3.218 | 3.028 | 14.217 |
| 19 | M | 14.38 | 21.93 | 15.36 | 37.291 | 12.75 | 12.36 | 12.08 | 16.59 | 9.14 | 8.03 | 7.27 | 41.03 |
| | sd | 18.887 | 5.347 | 3.478 | 8.519 | 5.148 | 4.977 | 5.133 | 5.83 | 4.106 | 3.258 | 3.141 | 15.002 |
| 20 | M | 15.22 | 22.24 | 15.41 | 37.65 | 13.3 | 12.85 | 12.74 | 16.84 | 9.51 | 8.1 | 7.46 | 41.91 |
| | sd | 19.878 | 5.383 | 3.516 | 8.63 | 5.551 | 5.464 | 5.574 | 6.068 | 4.283 | 3.323 | 3.266 | 15.776 |
| 21 | M | 16.58 | 21.952 | 15.29 | 37.24 | 13.57 | 12.96 | 13.001 | 16.741 | 9.551 | 8.1 | 7.5 | 41.89 |
| | sd | 21.943 | 5.551 | 3.613 | 8.889 | 5.509 | 5.436 | 5.589 | 5.993 | 4.361 | 3.291 | 3.197 | 15.692 |
| Total | M | 14.75 | 22.033 | 15.42 | 37.45 | 12.801 | 12.36 | 12.1 | 16.522 | 8.983 | 7.98 | 7.24 | 40.72 |
| | sd | 19.132 | 5.362 | 3.486 | 8.548 | 5.151 | 4.997 | 5.153 | 5.865 | 4.125 | 3.267 | 3.15 | 15.047 |
| | η² | 0.002 | 0.001 | 0.001 | 0.001 | 0.009 | 0.008 | 0.014 | 0.002 | 0.018 | 0.002 | 0.005 | 0.006 |
| | F | 6.956 | 3.858 | 3.93 | 3.34 | 36.485 | 31.974 | 56.214 | 9.042 | 73.323 | 7.93 | 19.103 | 24.509 |
| | P | <0.001 | 0.004 | 0.003 | 0.01 | <0.001 | <0.001 | <0.001 | <0.001 | <0.001 | <0.001 | <0.001 | <0.001 |
| **Grade** | | | | | | | | | | | | | |
| Freshman | M | 14.54 | 22.28 | 15.71 | 37.99 | 12.142 | 11.73 | 11.26 | 15.751 | 7.97 | 7.571 | 6.74 | 38.04 |
| | sd | 17.93 | 5.161 | 3.338 | 8.17 | 4.596 | 4.341 | 4.503 | 5.673 | 3.721 | 3.224 | 3.009 | 14.02 |
| Sophomores | M | 14.46 | 22.052 | 15.35 | 37.4 | 13.15 | 12.77 | 12.6 | 17 | 9.63 | 8.232 | 7.551 | 42.41 |
| | sd | 19.113 | 5.419 | 3.51 | 8.645 | 5.433 | 5.325 | 5.457 | 5.947 | 4.244 | 3.269 | 3.203 | 15.478 |
| Juniors | M | 16.43 | 21.661 | 15.12 | 36.78 | 13.001 | 12.39 | 12.32 | 16.68 | 9.16 | 8.061 | 7.322 | 41.22 |
| | sd | 21.752 | 5.484 | 3.625 | 8.824 | 5.341 | 5.242 | 5.314 | 5.915 | 4.197 | 3.309 | 3.201 | 15.245 |
| Seniors | M | 14.38 | 20.701 | 14.511 | 35.21 | 13.8 | 13.02 | 13.1 | 17.14 | 9.62 | 8.302 | 7.631 | 42.68 |
| | sd | 20.301 | 5.817 | 3.777 | 9.299 | 5.092 | 4.977 | 5.286 | 5.66 | 4.104 | 3.109 | 3.002 | 14.534 |
| Total | M | 14.75 | 22.03 | 15.42 | 37.453 | 12.8 | 12.36 | 12.1 | 16.52 | 8.98 | 7.981 | 7.241 | 40.72 |
| | sd | 19.132 | 5.362 | 3.486 | 8.548 | 5.151 | 4.997 | 5.153 | 5.865 | 4.125 | 3.267 | 3.15 | 15.047 |

*(Continued)*

# Table 3. (Continued)

| | | Physical Exercise | Self-Efficacy | | | Negative Emotion | | | Smartphone Addiction | | | | |
|---|---|---|---|---|---|---|---|---|---|---|---|---|---|
| | | | Situational motivation | Subjective support | Total score | Pressure | Anxiety | Depression | Withdrawal symptoms | Highlight behaviour | Social comfort | Mood changes | Total score |
| | η² | 0.001 | 0.004 | 0.006 | 0.005 | 0.01 | 0.01 | 0.016 | 0.01 | 0.034 | 0.009 | 0.014 | 0.018 |
| | F | 6.17 | 18.721 | 32.103 | 24.605 | 50.187 | 50.434 | 82.732 | 51.76 | 185.327 | 45.662 | 75.254 | 96.213 |
| | P | <0.001 | <0.001 | <0.001 | <0.001 | <0.001 | <0.001 | <0.001 | <0.001 | <0.001 | <0.001 | <0.001 | <0.001 |

Male students demonstrated significantly higher levels of physical exercise and self-efficacy than female students; Seniors scored highest on smartphone addiction and negative emotions; regional differences were small but statistically significant.

**Table 4. The correlation between physical exercise, self-efficacy, smartphone addiction and negative emotions.**

| | Physical Exercise | Self-Efficacy | Situational motivation | Subjective support | Smartphone Addiction | Withdrawal symptoms | Highlight behaviour | Mood changes | Social comfort | Pressure | Anxiety | Depression |
|---|---|---|---|---|---|---|---|---|---|---|---|---|
| Physical Exercise | 1 | | | | | | | | | | | |
| Self-Efficacy | 0.303** | 1 | | | | | | | | | | |
| Situational motivation | 0.291** | 0.980** | 1 | | | | | | | | | |
| Subjective support | 0.298** | 0.953** | 0.874** | 1 | | | | | | | | |
| Smartphone Addiction | −0.093** | −0.085** | −0.070** | −0.101** | 1 | | | | | | | |
| Withdrawal symptoms | −0.087** | −0.088** | −0.078** | −0.097** | 0.954** | 1 | | | | | | |
| Highlight behaviour | −0.065** | −0.043** | −0.023* | −0.070** | 0.919** | 0.821** | 1 | | | | | |
| Mood changes | −0.070** | −0.066** | −0.052** | −0.084** | 0.917** | 0.832** | 0.828** | 1 | | | | |
| Social comfort | −0.124* | −0.115** | −0.105** | −0.122** | 0.866** | 0.775** | 0.713** | 0.737** | 1 | | | |
| Pressure | −0.046** | 0.021* | 0.033** | 0.001 | 0.540** | 0.478** | 0.532** | 0.508** | 0.476** | 1 | | |
| Anxiety | −0.057** | 0.031** | 0.045** | 0.007 | 0.540** | 0.474** | 0.539** | 0.510** | 0.473** | 0.938** | 1 | |
| Depression | −0.047** | 0.025* | 0.041** | 0 | 0.535** | 0.462** | 0.548** | 0.499** | 0.469** | 0.933** | 0.930** | 1 |

*The correlation is significant at the 0.01 level (two-tailed); Physical exercise showed a significant positive correlation with self-efficacy (r = 0.303, p < 0.01), and a significant negative correlation with smartphone addiction and negative emotions (r = −0.093 to −0.124, p < 0.01).

effect of self-efficacy and smartphone addiction on the relationship between physical exercise and pressure had a 95% CI of [0.0001, 0.0002], indicating a significant sequential mediation effect. Significant sequential mediation was also observed in the relationships between physical exercise and anxiety (95% CI: [0.001, 0.002]) and between physical exercise and depression (95% CI: [0.001, 0.002]). The mediation model of this study is illustrated in **Table 6**. The effect sizes for physical exercise on self-efficacy, smartphone addiction, and negative emotion were 0.815, −0.016, and −0.019, respectively. All P-values in the above analyses were less than 0.001, indicating statistical significance.

The mediation effects of this study are illustrated in **Fig 2**. The effect sizes of physical exercise were 0.815 on self-efficacy, −0.016 on smartphone addiction, and −0.019 on negative emotion. Self-efficacy, in turn, had an effect size of 0.0050 on negative emotion and 0.027 on smartphone addiction. All P-values in the analyses were below 0.001, indicating statistical significance.

**Table 5. Regression analysis results for physical exercise, self-efficacy, and smartphone addiction on negative emotions (controlling for gender, year group, region, and age).**

| Regression | | Fitting indices | | | Coefficient | | |
|---|---|---|---|---|---|---|---|
| Outcome variables | Predictive variables | R | R² | F | β | SE | t |
| Pressure | | 0.562 | 0.315 | 365.653*** | | | |
| | Self-Efficacy | | | | 0.041 | 0.004 | 9.581 |
| | Situational motivation | | | | 0.185 | 0.002 | 80.501 |
| | Physical Exercise | | | | −0.156 | 0.002 | −7.884 |
| | region | | | | −0.712 | 0.71 | −2.89 |
| | grade | | | | 0.111 | 0.57 | 1.948 |
| | gender | | | | −1.219 | 0.76 | −16.047 |
| | age | | | | 0.179 | 0.042 | 4.261 |
| Anxiety | | 0.564 | 0.318 | 1044.468*** | | | |
| | Self-Efficacy | | | | 0.05 | 0.004 | 12.088 |
| | Situational motivation | | | | 0.179 | 0.002 | 80.906 |
| | Physical Exercise | | | | −0.019 | 0.002 | 2.601 |
| | region | | | | −0.83 | 0.069 | −1.207 |
| | grade | | | | 0.0478 | 0.055 | 0.866 |
| | gender | | | | −1.167 | 0.736 | −15.857 |
| | age | | | | 0.176 | 0.406 | 4.322 |
| Depression | | 0.566 | 0.321 | 1059.345*** | | | |
| | Self-Efficacy | | | | 0.038 | 0.004 | 9.072 |
| | Situational motivation | | | | 0.182 | 0.002 | 79.747 |
| | Physical Exercise | | | | −0.196 | 0.002 | −9.989 |
| | region | | | | −0.215 | 0.071 | −3.042 |
| | grade | | | | 0.197 | 0.057 | 3.469 |
| | gender | | | | −1.538 | 0.078 | −20.308 |
| | age | | | | 0.235 | 0.042 | 5.627 |
| Situational motivation | | 0.1512 | 0.229 | 73.387*** | | | |
| | Physical Exercise | | | | −0.092 | 0.007 | −13.872 |
| | Region | | | | 0.774 | 0.248 | 3.127 |
| | grade | | | | 1.918 | 0.198 | 9.696 |
| | gender | | | | −1.531 | 0.264 | −5.805 |
| | age | | | | 0.188 | 0.146 | 1.285 |
| Self-Efficacy | | 0.35 | 0.123 | 365.653*** | | | |
| | Physical Exercise | | | | 0.124 | 0.004 | 34.583*** |
| | Situational motivation | | | | −0.038 | 0.004 | −8.850*** |
| | region | | | | 0.885 | 0.133 | 6.642 |
| | grade | | | | −0.855 | 0.107 | −8.005 |
| | gender | | | | −1.712 | 0.142 | −12.05 |
| | age | | | | 0.352 | 0.079 | 4.467 |

***p<0.001, **p<0.01; Physical exercise significantly negatively predicts smartphone addiction and negative emotions; Self-efficacy positively predicts negative emotions; Smartphone addiction positively predicts negative emotions.

**Table 6. Analysis of the mediating effect of self-efficacy and smartphone addiction on the relationship between physical exercise and negative emotions (bootstrap = 5000).**

| Variable | Name | Effect | Boot SE | LLCI | ULCI |
|---|---|---|---|---|---|
| Pressure | Total effect | −0.027 | 0.002 | −0.031 | −0.023 |
| | Direct effect | −0.016 | 0.002 | −0.019 | −0.012 |
| | Indirect effect | −0.012 | 0.002 | −0.015 | −0.009 |
| | Physical Exercise→Smartphone Addiction→Pressure | −0.017 | 0.001 | −0.019 | −0.014 |
| | Physical Exercise→Self-Efficacy→Pressure | 0.005 | 0.001 | 0.004 | 0.006 |
| | Physical Exercise→Smartphone Addiction→Self-Efficacy→Pressure | −0.001 | 0.001 | 0.001 | 0.002 |
| Anxiety | Total effect | −0.029 | 0.002 | −0.034 | −0.025 |
| | Direct effect | −0.019 | 0.002 | −0.023 | −0.015 |
| | Indirect effect | −0.012 | 0.002 | −0.013 | −0.007 |
| | Physical Exercise→Smartphone Addiction→Anxiety | −0.017 | 0.001 | −0.019 | −0.014 |
| | Physical Exercise→Self-Efficacy→Anxiety | 0.005 | 0.001 | 0.005 | 0.007 |
| | Physical Exercise→Smartphone Addiction→Self-Efficacy→Anxiety | 0.001 | 0.001 | 0.001 | 0.002 |
| Depression | Total effect | −0.031 | 0.002 | −0.036 | −0.027 |
| | Direct effect | −0.019 | 0.002 | −0.024 | −0.016 |
| | Indirect effect | −0.012 | 0.001 | −0.015 | −0.009 |
| | Physical Exercise→Smartphone Addiction→Depression | −0.017 | 0.001 | −0.019 | −0.014 |
| | Physical Exercise→Self-Efficacy→Depression | 0.005 | 0.001 | 0.004 | 0.006 |
| | Physical Exercise→Smartphone Addiction→Self-Efficacy→Depression | 0.001 | 0.001 | 0.001 | 0.002 |

Smartphone addiction and self-efficacy both exert significant mediating effects between physical exercise and negative emotions, with the chain mediation effect being statistically significant (95% confidence interval does not include zero).

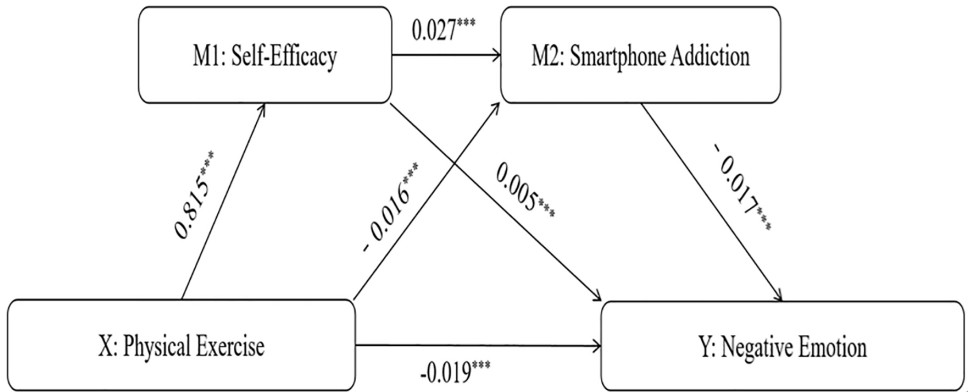

**Fig 2. A chain-mediated pathway model of physical exercise influencing negative emotions via self-efficacy and smartphone addiction.** Note: ***$p < 0.001$, **$p < 0.01$; Path coefficients indicate that physical exercise indirectly reduces negative emotions (β = −0.019) by enhancing self-efficacy (β = −0.815) and decreasing smartphone addiction (β = −0.016).

## Discussion

This study provides an in-depth exploration of the mechanisms through which college students' physical exercise affects negative emotion, with a particular focus on the mediating roles of self-efficacy and smartphone addiction. The findings

validate the following hypotheses: *H1* - physical exercise significantly negatively predicts negative emotion; *H2* - self-efficacy mediates the relationship between physical exercise and negative emotion; *H3* - smartphone addiction similarly mediates the relationship between physical exercise and negative emotion; and *H4* - self-efficacy and smartphone addiction form a chain mediation effect between physical exercise and negative emotion. The empirical findings indicate that physical exercise not only directly alleviates negative emotions, but also exerts an indirect effect through a chain-mediating pathway involving enhanced self-efficacy and reduced smartphone addiction, a mechanism supported by the frameworks of SCT and SDT. Furthermore, the study reveals that as the frequency and intensity of physical exercise increase among college students, their self-efficacy improves significantly, and their levels of smartphone addiction decrease markedly, ultimately leading to a substantial alleviation of negative emotion. These results demonstrate that physical exercise not only enhances college students' self-efficacy but also effectively reduces their dependence on smartphones, thereby significantly mitigating negative emotions such as anxiety and depression. The study presents new scientific evidence for promoting the mental health of college students, further underscoring the multifaceted role of physical exercise in enhancing psychological well-being and its underlying mechanisms. The results highlight the complex influence of physical exercise on the mental health of university students, emphasizing not only the direct benefits of exercise, but also its indirect effects on improving emotional health through two key mediating pathways: enhancing self-efficacy and reducing smartphone addiction. The finding of the chained mediation effect (self-efficacy and smartphone addiction) is particularly important, as it suggests that interventions targeting both aspects simultaneously may lead to more significant and lasting improvements in the psychological well-being of university students. These findings hold significant implications for universities and policymakers. Higher education institutions should increase investment in sports facilities, constructing more sports fields and gymnasiums, broadening the range of physical education courses, offering various forms of physical activity classes such as yoga and dance, and organising a diverse array of sporting events to encourage students to actively participate in physical exercise. Our research findings suggest that enhancing university students' self-efficacy should be prioritised. Universities could introduce mental health courses to educate students about mental well-being, provide counselling services, offer personalised psychological guidance, and encourage peer support to help students cultivate positive self-perceptions and enhance their capacity to cope with stress. Furthermore, addressing smartphone addiction, universities and policymakers should establish reasonable mobile phone usage guidelines, such as prohibiting mobile phone use in classrooms, designating "phone-free zones" in libraries and other locations, promoting digital health education to make students aware of the harms of excessive mobile phone use, and providing support for addiction recovery, including psychological counselling and support groups, to help students use mobile phones responsibly and mitigate their negative impacts. Universities should also integrate multiple intervention strategies, constructing a "physical-psychological-digital" three-pronged integrated intervention model to achieve optimal intervention outcomes.

## The predictive role of self-efficacy on negative emotion

First, the regression analysis in this study reveals a close interrelationship among physical exercise, self-efficacy, and negative emotions. Specifically, physical exercise significantly inhibits negative emotion, with self-efficacy acting as a crucial mediator. This finding is consistent with the results of the correlation analyses and regression models conducted in this study, indicating that self-efficacy is a crucial psychological mechanism through which physical exercise influences emotional well-being. This finding aligns with previous research, which has also highlighted the mediating role of self-efficacy in the relationship between physical activity and negative emotions, as well as the efficacy of physical exercise in reducing these emotions [61]. Second, the anticipation of successful emotion regulation can reinforce actual success. Individuals who anticipate successful emotion regulation through physical exercise tend to experience fewer negative emotions and are less prone to depression. According to self-efficacy theory, engaging in physical exercise enhances an individual's self-efficacy, promoting optimism about their ability to succeed and confidence in their capacity to regulate emotions through exercise, thereby mitigating negative emotion [62]. These findings are consistent with the results of the

regression analysis in this study. The underlying mechanisms may involve physical exercise directly influencing emotional, cognitive, neurophysiological, and neurochemical pathways, including the release of dopamine and endorphins, key neurotransmitters that regulate mood and reward. Furthermore, regular physical exercise can increase the expression of neurotrophic factors in the brain and reduce stress-induced neuroinflammatory responses, thus counteracting the propagation of negative emotion [2]. Moreover, college students with higher self-efficacy tend to report greater subjective well-being [42]. Similar findings have been reported in Western studies, demonstrating a negative correlation between physical exercise and negative affect, with self-efficacy acting as a mediator. However, the conceptualisation and manifestation of self-efficacy may differ across cultural contexts. In Western cultures, which often emphasise individualism, self-efficacy may be more strongly linked to individual independence and autonomy. Conversely, in Chinese culture, which tends to prioritise collectivism, self-efficacy may be more closely tied to an individual's contribution to collective goals and their sense of responsibility towards others.

However, the observed positive correlation between physical exercise and negative emotion in certain instances deviates from previous research findings. This study proposes that this phenomenon may arise because some individuals utilize physical exercise as a means of venting negative emotions. Faced with negative emotions, such individuals may consciously engage in physical exercise to release and alleviate their distress, thereby increasing their self-efficacy. Consequently, the more they exercise, the higher their self-efficacy becomes—yet their underlying negative emotions may persist or even intensify, potentially leading to a positive correlation between physical exercise and negative emotion. This observation may be related to the characteristics of the sample, with some students using physical exercise as a coping mechanism for negative affect. Similar phenomena have been observed in Western cultures, with some individuals utilising exercise to manage stress. However, the specific coping mechanisms employed may vary across different cultural contexts. In China, for instance, students may experience greater levels of negative affect due to academic pressures and societal expectations [63], potentially leading to a more frequent use of physical exercise as a coping strategy. It is important to note that the effectiveness of using physical exercise as a means of emotional release may depend on the type, intensity, and duration of the exercise [64]. If the exercise is not appropriate or is excessive, it may lead to physical fatigue, injury, and even exacerbate negative emotions [65]. Therefore, within the intervention, students should be guided to adopt scientific and appropriate exercise methods, and combined with other psychological support measures, to ensure the positive effects of physical exercise.

### The mediating role of self-efficacy

According to Bandura's SCT [66], when college students engage in physical exercise and receive support and encouragement from their environment, their self-efficacy may be further enhanced. Exercise self-efficacy, in particular, plays a crucial role in determining individuals' activity choices and the level of effort they invest [67]. It influences an individual's persistence in activities and shapes their attitudes towards challenges, as well as the acquisition and execution of new behaviours. In this study, the mediating effect of self-efficacy between physical exercise and negative emotions was significant, validating the applicability of this theory within the university student population and highlighting the enhancement of self-efficacy as a key pathway through which physical exercise improves emotional well-being. Individuals with strong self-efficacy tend to pursue challenging goals with greater confidence in their abilities. Conversely, those with low self-efficacy often doubt their capabilities and worry about their competence when facing failures or setbacks [27]. Another study highlights that self-efficacy has a significant impact on mental health, often leading to increased motivation in individuals.

Therefore, physical exercise may enhance self-discipline, such as maintaining regular and consistent exercise routines. As individuals progressively improve their capabilities, their self-efficacy is subsequently strengthened. The continuous physical improvement and sense of achievement derived from completing exercise tasks further elevate self-efficacy. This suggests that as college students increase their level of physical exercise, they typically gain greater self-confidence, which in turn leads to improved functional capacity and stronger self-efficacy.

## The mediating role of smartphone addiction

The 51st Statistical Report on Internet Development in China, released by CNNIC [32], reveals that smartphone addiction is a particularly significant issue among Chinese college students, necessitating urgent intervention. This phenomenon has the potential to impact both physical and mental health negatively, potentially leading to insomnia, depression, and anxiety [35,36], and ultimately exacerbating negative emotions. In this study, smartphone addiction was significantly and positively correlated with negative emotions, and the mediating effect was also significant, indicating that reducing smartphone addiction is a vital strategy for mitigating negative emotions. This finding is also consistent with the tenets of SDT, which posits that excessive smartphone use may undermine an individual's sense of autonomy, competence, and relatedness, thereby impairing psychological well-being. Research has shown that physical exercise can directly reduce the frequency of smartphone use among college students [21] and alleviate their daily negative emotions [22], suggesting that increased physical activity could offer a viable solution. This study further demonstrates that smartphone addiction correlates negatively with individuals' physical and mental health, likely due to excessive smartphone use displacing time allocated for physical exercise and overloading users with fragmented information, thus contributing to the development of negative emotions. Compared to studies conducted in Western contexts, the issue of smartphone addiction among Chinese university students may be more pronounced. This could be attributed to the intense competitive environment within Chinese society, the pressures of academic performance, and a greater reliance on social media platforms. Whilst smartphone addiction is also a recognised concern in the West, its manifestations and contributing factors may differ. Western students may be more inclined to use smartphones for entertainment purposes, whereas Chinese students may utilise them more frequently for both academic work and social interaction [68].

Consequently, physical exercise may help individuals reduce their propensity for smartphone addiction by replacing online time with physical activity. In addition to mitigating smartphone addiction, this shift simultaneously enhances physical fitness and enriches daily life. This implies that individuals tending to smartphone addiction often sacrifice time that could be spent on physical exercise, highlighting the potential of increased physical activity to alleviate smartphone addiction effectively.

## The chain mediating role of self-efficacy and smartphone addiction in the impact of physical exercise on negative emotions

According to SDT [52], physical exercise can fulfil an individual's fundamental psychological needs, encompassing autonomy, competence, and relatedness. When college students experience a sense of control over their actions (autonomy) and enhanced ability (competence) through physical exercise, their self-efficacy is significantly boosted [69]. Individuals with high self-efficacy are more likely to self-regulate their behaviour, potentially mitigating smartphone addiction [70]. Moreover, the relationship between smartphone addiction and negative emotions can be explained through SDT's frustration mechanism of basic psychological needs: excessive smartphone use can weaken autonomy and relatedness, impeding the fulfilment of psychological needs and consequently exacerbating negative emotions [71].The chain-mediating model validated in this study integrated perspectives from both SCT and SDT, demonstrating that physical exercise ultimately reduces negative emotions via a dual mechanism: enhancing self-efficacy and reducing smartphone addiction. These theoretical underpinnings align with the findings of this study, further supporting the chain mediation effect, whereby physical exercise reduces negative emotions by enhancing self-efficacy, which subsequently alleviates smartphone addiction.

The chained mediation effect observed in this study indicates that the relationship between physical exercise and negative emotions is not a straightforward linear one, but rather a complex and multi-faceted process. By enhancing students' self-efficacy and assisting them in reducing their reliance on smartphones, a positive cycle can be established, thereby more effectively improving their psychological well-being. The findings of this study indicate that college students

with strong physical exercise habits are less likely to experience negative emotions [72]. These students utilize physical exercise as a means of self-regulation. As their capacity for physical exercise improves, their self-efficacy also increases, driven by enhanced competence. This, in turn, significantly reduces the likelihood of smartphone addiction, thereby mitigating negative emotions [73]. Therefore, increased physical exercise effectively reduces negative emotions through a chain pathway: boosting self-efficacy and alleviating smartphone addiction, which ultimately safeguards the mental and physical well-being of college students.

## Implications and recommendations for mental health interventions

Based on the empirical findings presented, this study offers significant and detailed implications for the practical implementation of mental health interventions in university students. For this population, interventions should extend beyond simply encouraging physical exercise, employing structured programmes such as "exercise prescriptions" and integrating these within the campus mental health service system. These programmes should be explicitly tailored to address the mediating pathways revealed in this study. Firstly, by establishing progressive, achievable exercise goals, providing peer encouragement and coaching feedback, and systematically building students' experiences of success, the intervention should aim to tangibly enhance their self-efficacy (corresponding pathway: physical exercise→self-efficacy). Personalised exercise plans should be developed, employing a "prescription" approach that considers students' physical condition, interests, and levels of negative emotion. For students experiencing higher levels of anxiety, relaxing exercises such as yoga and meditation are recommended; for those with higher levels of depression, aerobic exercises like running and swimming are advised. Furthermore, students should be encouraged to support one another in achieving their exercise goals, and regular self-efficacy assessments should be conducted to monitor progress and facilitate timely adjustments to the intervention plan. Secondly, a "digital wellbeing" module should be designed, implementing activities such as "exercise challenges" and "phone-free study rooms," to guide students to adopt aerobic exercise, team sports, and other activities as alternative rewards, thereby proactively reducing smartphone use (corresponding pathways: physical exercise→smartphone addiction; self-efficacy→smartphone addiction). Secondly, it is recommended to construct a "physical-psychological-digital" three-pronged integrated intervention model, combining physical exercise, cognitive behavioural therapy (CBT) – to enhance self-efficacy – and digital detoxification training, delivered collaboratively by counsellors, physical education instructors, and psychological advisors. This collaborative approach would gradually target multiple links within the chain-mediating model, thus maximising intervention effectiveness. Furthermore, when implementing intervention strategies, the unique characteristics of the Chinese cultural context should be taken into consideration. This could involve incorporating traditional Chinese health concepts, such as "yin and yang balance" and "harmony between humans and nature," to create intervention programmes that are more culturally relevant and appealing to Chinese students. It is also crucial to acknowledge the influence of the family, school, and broader social environment on students' mental well-being, and to integrate these factors into the development of more effective intervention approaches. Finally, to further enhance the effectiveness of the intervention, it is recommended to integrate behaviour change theories (such as the Theory of Planned Behaviour) into the design of the intervention programme. These theories can help us better understand the factors that influence individual behaviour change, thereby enabling more targeted interventions. This can be achieved by fostering a more positive attitude towards physical exercise and limiting smartphone use among students, and by enhancing their sense of control and self-efficacy, thereby promoting behavioural change.

## Research limitations and future research directions

This study has several limitations that warrant consideration for future research and expansion. (1) The cross-sectional design employed in this study, while revealing associations between variables, precludes the establishment of causal relationships. Future research should adopt longitudinal or experimental designs to further validate the causal pathways

between physical exercise, self-efficacy, smartphone addiction, and negative emotions. (2) The data relied on self-report questionnaires, which may be susceptible to recall bias and social desirability bias, thereby affecting the accuracy of the results. Furthermore, this study did not employ objective measurement tools (e.g., accelerometers, heart rate monitors, or smartphone screen time recording) to assess the intensity of physical exercise and smartphone usage. This may have led to discrepancies between subjective reports and actual behaviours, potentially affecting the precise estimation of the relationships between variables. Future studies could employ multi-method and multi-source data collection strategies, such as utilising objective monitoring devices like accelerometers and heart rate monitors to record physical exercise data, and leveraging the "digital wellbeing" features built into mobile phones or dedicated monitoring software to objectively obtain screen time data. This approach would help to mitigate the limitations of self-report measures and enhance the validity of the research. (3) The sample was limited to university students in Henan and Jiangsu provinces in China, and the specific cultural background and educational environment may restrict the generalisability of the findings. Future research should broaden the geographical sampling scope to encompass regions with varying levels of economic development and cultural characteristics within China, and cross-cultural comparative studies could be conducted to examine the generalisability and specificity of the chain-mediating model across university student populations from different cultural backgrounds. Moreover, future research should address the impact of multicollinearity on the study results, which can be done by increasing the independence of variables or employing more advanced statistical methods, such as partial least squares regression analysis. Furthermore, future research could explore the moderating effects of other potential variables (e.g., psychological resilience, social support, and personality traits) to provide a more comprehensive understanding of the boundary conditions under which this mechanism operates.

## Conclusions

The findings demonstrate a chain mediation effect whereby increased physical exercise among college students enhances self-efficacy, which in turn alleviates smartphone addiction, ultimately leading to a reduction in negative emotions. This discovery provides empirical evidence for the improvement of emotional well-being through physical exercise, particularly in the context of widespread smartphone use and heightened pressure among college students, underscoring the crucial role of physical exercise in mental health interventions.

## Acknowledgments

This study is a thank you to all the Chinese university students who participated in the questionnaire survey.

## Author contributions

**Conceptualization:** Rui Li, Hao-yu Li, Hao-jie Zuo, Mohamad Nizam Nazarudin.

**Data curation:** Jia-meng Sun, Bo Li.

**Formal analysis:** Chen-xi Li, Ning Zhou.

**Methodology:** Jia-meng Sun.

**Validation:** Qi Liu, Bo Li.

**Writing – original draft:** Rui Li, Hao-yu Li, Hao-jie Zuo, Mohamad Nizam Nazarudin.

**Writing – review & editing:** Rui Li, Hao-yu Li, Hao-jie Zuo, Mohamad Nizam Nazarudin.

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
