## [Decision Letter · Decision Letter 0]

13 Aug 2025

Dear Dr. Nazarudin,

Thank you for submitting your manuscript to PLOS ONE. After careful consideration, we feel that it has merit but does not fully meet PLOS ONE’s publication criteria as it currently stands. Therefore, we invite you to submit a revised version of the manuscript that addresses the points raised during the review process.

We look forward to receiving your revised manuscript.

Kind regards,

Leona Cilar Budler

Academic Editor

PLOS ONE

Journal Requirements:

3. Please include a complete copy of PLOS’ questionnaire on inclusivity in global research in your revised manuscript. Our policy for research in this area aims to improve transparency in the reporting of research performed outside of researchers’ own country or community. The policy applies to researchers who have travelled to a different country to conduct research, research with Indigenous populations or their lands, and research on cultural artefacts. The questionnaire can also be requested at the journal’s discretion for any other submissions, even if these conditions are not met.  Please find more information on the policy and a link to download a blank copy of the questionnaire here: https://journals.plos.org/plosone/s/best-practices-in-research-reporting. Please upload a completed version of your questionnaire as Supporting Information when you resubmit your manuscript.

4. In the online submission form, you indicated that If you require the corresponding data, please contact the corresponding author.

2025 Major Project of Philosophy and Social Science Research in Colleges and Universities: Research on the Formation Mechanism and Exercise Intervention of College Students' Mental Health Problems in the Era of Digital Intelligence

6. Please note that funding information should not appear in any section or other areas of your manuscript. We will only publish funding information present in the Funding Statement section of the online submission form. Please remove any funding-related text from the manuscript.

7. Please amend either the title on the online submission form (via Edit Submission) or the title in the manuscript so that they are identical.

8. Please include a separate caption for each figure in your manuscript.

9. Please ensure that you refer to Figure 1 in your text as, if accepted, production will need this reference to link the reader to the figure.

10. We note you have included a table to which you do not refer in the text of your manuscript. Please ensure that you refer to Table 2 in your text; if accepted, production will need this reference to link the reader to the Table.

11. Please include captions for your Supporting Information files at the end of your manuscript, and update any in-text citations to match accordingly. Please see our Supporting Information guidelines for more information: http://journals.plos.org/plosone/s/supporting-information.

12. Please remove all personal information, ensure that the data shared are in accordance with participant consent, and re-upload a fully anonymized data set.

Additional Editor Comments :

there are minor issues listed by reviewers. Please read all coments and suggestions to revise your paper. Also, check alljorunal guidelines for paper preparation.

Reviewers' comments:

Reviewer's Responses to Questions

**Comments to the Author**

1. Is the manuscript technically sound, and do the data support the conclusions?

Reviewer #1: Yes

Reviewer #2: Yes

Reviewer #3: Yes

Reviewer #4: Yes

2. Has the statistical analysis been performed appropriately and rigorously?

Reviewer #1: Yes

Reviewer #2: Yes

Reviewer #3: Yes

Reviewer #4: Yes

3. Have the authors made all data underlying the findings in their manuscript fully available?

Reviewer #1: Yes

Reviewer #2: Yes

Reviewer #3: Yes

Reviewer #4: Yes

4. Is the manuscript presented in an intelligible fashion and written in standard English?

Reviewer #1: Yes

Reviewer #2: Yes

Reviewer #3: Yes

Reviewer #4: Yes

Reviewer #1: Overall Impression: The study explores a relevant topic, examining the relationship between physical exercise, negative emotions, self-efficacy, and smartphone addiction among college students. The use of a large sample size and a chain mediation model strengthens the findings. However, the manuscript could benefit from some revisions to enhance its clarity, depth, and impact.

Pros (Strengths):

• Relevant Topic: Addresses a significant issue affecting college students, with practical implications for mental health interventions.

• Large Sample Size: The study utilizes a large sample (n = 16,355), increasing the statistical power and generalizability of the findings.

• Chain Mediation Model: The use of a chain mediation model provides a more nuanced understanding of the relationships between the variables.

• Clear Hypotheses: The hypotheses are clearly stated and logically derived from the literature.

• Comprehensive Statistical Analysis: The study employs appropriate statistical techniques, including descriptive analysis, correlation analysis, regression analysis, and mediation analysis.

• Data Availability: The authors confirm that all data are fully available without restriction.

• Addresses a gap in research: The research sheds light on the intricate connections between physical activity, mental well-being, self-assurance, and reliance on smartphones among students. It addresses a void in existing knowledge by investigating how these aspects interplay to influence students' emotional experiences.

Cons (Weaknesses):

• Limited Scope: The study focuses solely on college students in China, which may limit the generalizability of the findings to other populations.

• Cross-Sectional Design: The cross-sectional design limits the ability to draw causal inferences.

• Reliance on Self-Report Data: The study relies on self-report measures, which may be subject to recall bias and social desirability bias.

• Lack of Theoretical Depth: While the study mentions Bandura's Social Cognitive Theory (SCT), it could benefit from a more in-depth discussion of the theoretical underpinnings of the relationships between the variables.

• Some Ambiguity in the Discussion: Some interpretations in the discussion section could be more clearly linked back to the empirical findings.

• Clarity Issues: Improve the readability and flow for better engagement.

• Methodological Depth: Give a more detailed explanation of the data collection and analysis methods to enhance transparency and reproducibility.

• Addressing Limitations: Clearly recognize the study's constraints to handle issues like generalizability.

Suggestions for Improvement:

1. Expand the Introduction:

- Provide a more detailed overview of the existing literature on the relationships between physical exercise, negative emotions, self-efficacy, and smartphone addiction.

- Clearly articulate the gaps in the literature that the current study aims to address.

- Consider adding a section on the specific cultural context of college students in China and how it may influence the relationships between the variables.

- Elaborate on the rationale for choosing Henan and Jiangsu provinces.

2. Strengthen the Theoretical Framework:

- Provide a more in-depth discussion of Bandura's Social Cognitive Theory (SCT) and how it applies to the relationships between the variables.

- Explain how SCT can help explain the mediating roles of self-efficacy and smartphone addiction.

- Consider discussing other relevant theories, such as the self-determination theory, to provide a more comprehensive theoretical framework.

3. Address the Limitations of the Study:

- Acknowledge the limitations of the cross-sectional design and the inability to draw causal inferences.

- Discuss the potential for recall bias and social desirability bias due to the use of self-report measures.

- Address the limitations of generalizability due to the focus on college students in China.

4. Refine the Discussion Section:

- Ensure that all interpretations in the discussion section are clearly linked back to the empirical findings.

- Provide a more nuanced discussion of the implications of the findings for mental health interventions.

- Consider discussing the potential for future research to address the limitations of the current study.

- Make stronger recommendations for interventions based on the results.

5. Formatting and Clarity:

- Ensure that the manuscript is free of grammatical errors and typos.

- Use clear and concise language throughout the manuscript.

- Follow the formatting guidelines of the target journal.

- Improving the flow and coherence of paragraphs to maintain reader engagement.

6. Figures and Tables:

- Ensure that all figures and tables are clearly labeled and easy to understand.

- Provide a brief description of the key findings in the figure and table captions.

- In Table 2, clarify the labels "Risk-free," "Medium risk," and "High risk."

7. Consider These Points:

- Clarify how the data was collected, ensuring the process is replicable.

- Discuss how multicollinearity was addressed, especially given the high correlations among some variables.

- Explore whether the relationships found differ significantly across demographic groups (e.g., gender, region).

- Discuss the broader implications for universities and policymakers.

- Offer practical guidance for designing effective interventions based on study findings.

By addressing these weaknesses and incorporating these suggestions, it can enhance the quality and impact of the manuscript.

Reviewer #2: 1.The introduction section lacks a thorough review of previous research. The authors should conduct a more comprehensive analysis of existing studies, including their strengths and limitations, while clearly articulating the innovative aspects of this study and how it differs from prior research in the field.

2.Given that this study employs a cross-sectional design, special attention should be paid to the wording of hypotheses to avoid implying causal relationships. The language should be carefully framed to describe associations rather than causation throughout the manuscript.

3.The study's exclusive focus on Eastern and Central China may limit the generalizability of its findings. This geographical limitation should be explicitly acknowledged and discussed in the study's limitations section.

4.The discussion would benefit from comparative analysis with similar studies conducted in Western populations, as smartphone addiction patterns may vary due to cultural differences and other factors. We recommend adding a dedicated discussion on potential cultural influences, such as the possibility that Chinese students face greater academic pressure leading to different coping mechanisms.

5.While the conclusion appropriately notes the mental health benefits of physical exercise, it fails to provide practical implementation guidance. The authors should include specific recommendations for intervention strategies at the institutional level, such as how universities might practically apply these findings.

6.The reference formatting requires standardization, as some entries include DOI identifiers while others do not. We recommend ensuring all references follow a consistent format in accordance with the journal's style guidelines.

Reviewer #3: 1�Introduction Enhancement:

While the introduction provides context, expand the literature review to engage with prior studies—particularly those examining specific relationships between physical exercise, self-efficacy, and smartphone addiction—and integrate critical perspectives.

2�Theoretical Justification:

Hypotheses require stronger grounding in theoretical models and existing literature. Justify the selection of self-efficacy and smartphone addiction as mediators with explicit theoretical rationale.

3�Psychometric Validation:

Reliability and validity of scales (e.g., MPATS, DASS-21) are insufficiently addressed. Report internal consistency and validity metrics specific to this study’s sample.

4�Mediation Analysis Transparency:

Clarify procedural steps for mediation testing, including model selection criteria and bootstrap implementation methodology.

5�Discussion Depth:

Strengthen interpretation of chain mediation findings and articulate practical implications for university mental health interventions.

6�Design Limitations:

Explicitly acknowledge constraints of the cross-sectional design.

Reviewer #4: This manuscript presents a meaningful study with a respectable sample size and a well-structured analysis. However, there are still some issues that need to be revised.

1、Methods section: While the Physical Activity Rating Scale and other measurement tools demonstrated good internal consistency, the study does not provide detailed information regarding the validation of their psychometric properties, within the current sample.

2、Methods section: The study’s sample, though large, shows an uneven distribution of participants across gender and grade levels. This imbalance may affect the generalizability of the findings. The authors are advised to discuss the potential impact of this sampling bias and clarify whether stratified analysis or weighting methods were employed to address these discrepancies.

3、Methods section: The study relies on self-reported measures for key variables, which may introduce recall or social desirability biases. The authors should elaborate on steps taken to mitigate these biases and discuss the limitations of self-reported data in the context of their findings.

4、Discussion section: The study employs a cross-sectional design, which inherently limits the ability to establish causal relationships between variables. While the chain mediation model is compelling, the authors should explicitly acknowledge this limitation and recommend longitudinal or experimental designs for future research to validate causality.

5、Discussion section: While the results are thoroughly presented, the discussion could benefit from a deeper theoretical integration.

6、Discussion section: The authors note that physical exercise may sometimes correlate positively with negative emotions due to emotional venting.

7、Limitations: The study mentions potential biases from questionnaire responses but does not address the lack of objective measures. Including these limitations would provide a more balanced perspective on the study’s constraints.

**Do you want your identity to be public for this peer review?** For information about this choice, including consent withdrawal, please see our Privacy Policy

Reviewer #1: No

Reviewer #2: No

Reviewer #3: No

Reviewer #4: No

---

## [Author Response · Author response to Decision Letter 1]

22 Oct 2025

Point-by-point Responses to Reviewer 1

Dear Editor and dear reviewers

Re: Manuscript Number: PONE-D-25-38262 and Title: The Impact of College Students' Physical Exercise on Negative Emotion: The Chain Mediating Role of Self-efficacy and Smartphone Addiction

Thank you very much for your comments and professional advice. These opinions help to improve the academic rigor of our manuscript. Based on your suggestion and request, we have made corrected modifications to the revised manuscript. Here are point-by-point responses to your comments. We hope that our work can be improved again. Furthermore, we would like to show the details as follows:

Sincerely,

Comment #1:

Expand the Introduction:

-Provide a more detailed overview of the existing literature on the relationships between physical exercise, negative emotions, self-efficacy, and smartphone addiction.

-Clearly articulate the gaps in the literature that the current study aims to address.

-Consider adding a section on the specific cultural context of college students in China and how it may influence the relationships between the variables.

-Elaborate on the rationale for choosing Henan and Jiangsu provinces.

Response#1:

Thank you for your valuable suggestions regarding the introduction. We have comprehensively revised the introduction in accordance with your recommendations. In response to the suggestion to ‘provide a more detailed overview of the existing literature,’ we have expanded the literature review on the relationship between physical exercise, negative emotions, self-efficacy, and smartphone addiction. Specific modifications can be found in lines 91-99 and 155-161 and 193-200 of the manuscript. To clarify the research objectives, we have explicitly stated the literature gap addressed by the present study, as detailed in lines 99-109 of the manuscript. Furthermore, we have incorporated a discussion on the cultural context of Chinese university students and its potential influence on the relationships between variables, as outlined in lines 224-232 of the manuscript. We extend our gratitude once more for your guidance. Additionally, we have elaborated on the rationale for selecting Henan and Jiangsu provinces, as explained in lines 233-248. We extend our gratitude once more for your guidance.

Comment #2:

Strengthen the Theoretical Framework:

-Provide a more in-depth discussion of Bandura's Social Cognitive Theory (SCT) and how it applies to the relationships between the variables.

-Explain how SCT can help explain the mediating roles of self-efficacy and smartphone addiction.

-Consider discussing other relevant theories, such as the self-determination theory, to provide a more comprehensive theoretical framework.

Response#2:

Thank you for your valuable suggestions regarding the theoretical framework. We have substantially refined the theoretical framework in the introduction as suggested. Addressing the recommendation to ‘discuss Bandura's Social Cognitive Theory (SCT) in greater depth and how it applies to the relationships between variables,’ we have elaborated on SCT's core concepts within the introduction and provided a detailed analysis of how SCT explains the relationship between physical exercise, negative emotions, self-efficacy, and smartphone addiction. Additionally, we have specifically clarified how SCT aids in explaining the mediating role of self-efficacy in smartphone addiction. Furthermore, to provide a more comprehensive theoretical framework, we have incorporated a discussion of Self-Determination Theory (SDT), exploring its complementarity with SCT and how both theories collectively account for the relationships among these variables. Please refer to lines 249-296 of the manuscript. We believe these modifications will further enhance the theoretical depth of the manuscript.

Comment #3:

Address the Limitations of the Study:

-Acknowledge the limitations of the cross-sectional design and the inability to draw causal inferences.

-Discuss the potential for recall bias and social desirability bias due to the use of self-report measures.

-Address the limitations of generalizability due to the focus on college students in China.

Response#3:

We are grateful for your detailed suggestions regarding the limitations of this study. We have incorporated your recommendations by providing a clear and comprehensive discussion of the study's limitations within the manuscript. We fully acknowledge the limitations inherent in a cross-sectional design and emphasise that this study cannot draw causal inferences. Furthermore, we discuss the potential for recall bias and social desirability bias arising from the use of self-report measures, analysing their possible impact on the findings in the Discussion section. Finally, we explicitly acknowledge the generalisability limitations arising from our focus on Chinese university students within the discussion section. We also propose strategies for future research to enhance the generalisability of findings. Please refer to lines 785-816 of the manuscript. We believe these supplementary clarifications will enhance transparency regarding the study's limitations and provide readers with a more accurate understanding.

Comment #4:

Refine the Discussion Section:

-Ensure that all interpretations in the discussion section are clearly linked back to the empirical findings.

-Provide a more nuanced discussion of the implications of the findings for mental health interventions.

-Consider discussing the potential for future research to address the limitations of the current study.

-Make stronger recommendations for interventions based on the results.

Response#4:

We are grateful for your valuable suggestions to enhance the discussion section of this study. We have carefully revised the discussion to strengthen its rigour and practical relevance. We have ensured all interpretations within the discussion are closely tied to empirical findings, with interpretations of the results grounded in objective data and statistical analysis. Specific revisions can be found in lines 556-559 and 604-6066 and 660-672 and 682-691 and 720-723 of the manuscript. Additionally, we have provided a more detailed discussion on the implications of our findings for mental health interventions, elaborating on how the results can be utilised to design and implement mental health interventions for university students, as detailed in lines 742-784 of the manuscript. Furthermore, we have explored the potential for future research to address current limitations and proposed specific avenues for further investigation to deepen understanding of related issues, as detailed in lines723-816 of the manuscript. Finally, we have formulated stronger intervention recommendations based on our findings, emphasising the potential benefits of these measures, as outlined in lines 742-784 of the manuscript. We believe these revisions will significantly enhance the depth and practical value of the discussion section.

Comment #5:

Formatting and Clarity:

-Ensure that the manuscript is free of grammatical errors and typos.

-Use clear and concise language throughout the manuscript.

-Follow the formatting guidelines of the target journal.

-Improving the flow and coherence of paragraphs to maintain reader engagement.

Response#5:

We are grateful for your detailed suggestions regarding the format and clarity of this study. We have undertaken a thorough and meticulous revision of the manuscript in accordance with your recommendations. The entire text has been carefully proofread to ensure the absence of grammatical errors and spelling mistakes, with particular attention paid to achieving clarity and conciseness in linguistic expression. Concurrently, we have strictly adhered to the target journal's formatting guidelines, adjusting the manuscript's layout to meet its requirements. Furthermore, we have enhanced paragraph flow and coherence to clarify the article's logical structure and improve readability, thereby better retaining reader engagement. For specific modifications, please refer to the full manuscript. We believe these improvements will render the paper more accessible and easier to comprehend.

Comment #6:

Figures and Tables:

-Ensure that all figures and tables are clearly labelled and easy to understand.

-Provide a brief description of the key findings in the figure and table captions.

-In Table 2, clarify the labels "Risk-free," "Medium risk," and "High risk."

Response#6:

Thank you for your specific suggestions regarding the figures and tables in this study. We have carefully reviewed and revised the figures and tables in the manuscript to ensure they are clear and easily understood. We have ensured that all relevant figures and tables are accompanied by clear labels, with their captions providing concise summaries of key findings to enable readers to grasp information swiftly. Please refer to all figures and tables within the manuscript for the specific modifications made. Regarding your suggestion concerning Table 2, the labels ‘Risk-free’, ‘medium risk’ and ‘high risk’ have been clarified; please see lines 898 -902 and 418-422 of the manuscript for details. We believe these amendments will enhance the readability of the figures and tables and improve the accuracy of the information conveyed.

Comment #7:

Consider These Points:

-Clarify how the data was collected, ensuring the process is replicable.

-Discuss how multicollinearity was addressed, especially given the high correlations among some variables.

-Explore whether the relationships found differ significantly across demographic groups (e.g., gender, region).

-Discuss the broader implications for universities and policymakers.

-Offer practical guidance for designing effective interventions based on study findings.

Response#7:

We are grateful for your additional suggestions regarding this study. We have carefully considered these comments and made further refinements to the manuscript. The specific procedures for data collection have been detailed in the Methods section to ensure the reproducibility of the process; please refer to lines 300-316 of the manuscript for the specific modifications. Regarding the issue of multicollinearity, we have provided corresponding explanations, detailed in lines 449-454 of the manuscript. Furthermore, to explore variations in the identified relationships across different demographic groups (e.g., gender, region), we conducted additional analyses, detailed in lines 485-503 of the manuscript. Additionally, we have discussed the broader implications of our findings for universities and policymakers in the Discussion section, offering corresponding recommendations, detailed in lines 573-599 of the manuscript. Finally, based on our findings, we provide practical guidance for designing effective interventions, as detailed in lines 742-764 of the manuscript. We believe these revisions will enhance the manuscript's completeness and practical utility.

Point-by-point Responses to Reviewer 2

Dear Editor and dear reviewers

Re: Manuscript Number: PONE-D-25-38262 and Title: The Impact of College Students' Physical Exercise on Negative Emotion: The Chain Mediating Role of Self-efficacy and Smartphone Addiction

Thank you very much for your comments and professional advice. These opinions help to improve the academic rigor of our manuscript. Based on your suggestion and request, we have made corrected modifications to the revised manuscript. Here are point-by-point responses to your comments. We hope that our work can be improved again. Furthermore, we would like to show the details as follows:

Sincerely,

Comment #1:

The introduction section lacks a thorough review of previous research. The authors should conduct a more comprehensive analysis of existing studies, including their strengths and limitations, while clearly articulating the innovative aspects of this study and how it differs from prior research in the field.

Response#1:

We appreciate your insightful comment regarding the introduction’s literature review. In response, we have significantly expanded the introduction to provide a more comprehensive analysis of previous research. This includes a more thorough examination of existing studies, highlighting both their strengths and limitations. We have also explicitly articulated the innovative aspects of our study, clarifying how it differs from prior research to the field. Please refer to lines 99-109 and 127-136 and 205-214 of the manuscript.

Comment #2:

Given that this study employs a cross-sectional design, special attention should be paid to the wording of hypotheses to avoid implying causal relationships. The language should be carefully framed to describe associations rather than causation throughout the manuscript.

Response#2:

Thank you for your valuable feedback on our research. We have carefully considered your comments regarding the study design and wording. In response to your concern about the cross-sectional design, we have meticulously reviewed and revised the wording of our hypotheses and the entire manuscript to ensure that our language accurately reflects the correlational nature of the study, avoiding any implication of causal relationships. We have focused on describing associations rather than causation throughout. Specific revisions can be found throughout the manuscript.

Comment #3:

The study's exclusive focus on Eastern and Central China may limit the generalizability of its findings. This geographical limitation should be explicitly acknowledged and discussed in the study's limitations section.

Response#3:

Thank you for your insightful comment. We acknowledge the reviewer’s concern regarding the geographical limitations of our study, specifically its focus on Eastern and Central China, and its potential impact on the generalizability of our findings. In response, we have explicitly acknowledged this limitation and discussed its implications within the limitations section of the manuscript. Please refer to the limitations section (lines 803-816) for details.

Comment #4:

The discussion would benefit from comparative analysis with similar studies conducted in Western populations, as smartphone addiction patterns may vary due to cultural differences and other factors. We recommend adding a dedicated discussion on potential cultural influences, such as the possibility that Chinese students face greater academic pressure leading to different coping mechanisms.

Response#4:

Thank you for your insightful feedback. We appreciate your suggestion to enhance the discussion section with a comparative analysis against studies conducted in Western populations. We recognize that smartphone addiction patterns and associated factors may vary due to cultural differences and other influences. Therefore, we have added a dedicated section to the discussion that specifically addresses potential cultural influences, including the possibility that Chinese students may experience greater academic pressure, potentially leading to different coping mechanisms and affecting the relationships between the variables studied. Please refer to lines 623-631 and 644-653 and 695-702 and 770-777 of the manuscript.

Comment #5:

While the conclusion appropriately notes the mental health benefits of physical exercise, it fails to provide practical implementation guidance. The authors should include specific recommendations for intervention strategies at the institutional level, such as how universities might practically apply these findings.

Response#5:

Thank you for your insightful comment on the conclusion. We acknowledge the reviewer’s suggestion to enhance the practical implications of our findings. In response, we have revised the conclusion to include specific recommendations for intervention strategies at the institutional level. These recommendations outline how universities could practically apply our findings to promote physical exercise and support students’ mental health. Please refer to lines 742-784 of the manuscript.

Comment #6:

The reference formatting requires standardization, as some entries include DOI identifiers while others do not. We recommend ensuring all references follow

---

## [Decision Letter · Decision Letter 1]

4 Nov 2025

Dear Dr. Nazarudin,

Thank you for submitting your manuscript to PLOS ONE. After careful consideration, we feel that it has merit but does not fully meet PLOS ONE’s publication criteria as it currently stands. Therefore, we invite you to submit a revised version of the manuscript that addresses the points raised during the review process.

We look forward to receiving your revised manuscript.

Kind regards,

Leona Cilar Budler

Academic Editor

PLOS ONE

Journal Requirements:

Additional Editor Comments:

Dear,

one reviewer still has doubts about the sampling method of this study, it does not affect the main conclusions of the research. Authors should resolve that.

Reviewers' comments:

Reviewer's Responses to Questions

**Comments to the Author**

Reviewer #2: All comments have been addressed

Reviewer #4: All comments have been addressed

2. Is the manuscript technically sound, and do the data support the conclusions?

Reviewer #2: Yes

Reviewer #4: Yes

3. Has the statistical analysis been performed appropriately and rigorously?

Reviewer #2: Yes

Reviewer #4: Yes

4. Have the authors made all data underlying the findings in their manuscript fully available?

Reviewer #2: Yes

Reviewer #4: Yes

5. Is the manuscript presented in an intelligible fashion and written in standard English?

Reviewer #2: Yes

Reviewer #4: Yes

Reviewer #2: The author's paper has been well revised and is recommended for publication.Although I still have some doubts about the sampling method of this study, it does not affect the main conclusions of the research.

Reviewer #4: This study has a large sample size, with relatively precise sampling, conforming to the mainstream paradigm of current scientific research. In particular, the selection of variables in this study is well-founded and thoroughly justified.

**Do you want your identity to be public for this peer review?** For information about this choice, including consent withdrawal, please see our Privacy Policy

Reviewer #2: No

Reviewer #4: No

---

## [Author Response · Author response to Decision Letter 2]

11 Nov 2025

Dear Editor and dear reviewers

Re: Manuscript Number: PONE-D-25-38262 and Title: The Impact of College Students' Physical Exercise on Negative Emotion: The Chain Mediating Role of Self-efficacy and Smartphone Addiction

Thank you very much for your comments and professional advice. These opinions help to improve the academic rigor of our manuscript. Based on your suggestion and request, we have made corrected modifications to the revised manuscript. Here are point-by-point responses to your comments. We hope that our work can be improved again. Furthermore, we would like to show the details as follows:

Sincerely,

Comment #1:

one reviewer still has doubts about the sampling method of this study; it does not affect the main conclusions of the research. Authors should resolve that.

Response#1:

We sincerely appreciate this valuable feedback from the reviewers. This study employed the “multistage stratified cluster sampling” method widely used in large-scale social surveys. The design aimed to maximize the representativeness of the sample for China's college student population within limited resources. The sampling process comprised three distinct stages: First, we purposefully selected Jiangsu Province in East China and Henan Province in Central China based on their representativeness in economic development and educational resources, seeking to encompass students from diverse socioeconomic backgrounds; Second, within each province, we randomly selected a total of 8 institutions encompassing both undergraduate and vocational college levels. Third, at each selected institution, we stratified by academic year and major, then randomly clustered classes as survey units. This design systematically accounts for multiple dimensions—region, institution type, academic year, and major—to construct a logically structured sample framework.

We fully acknowledge that despite this design, the final sample exhibits some imbalance in gender distribution (60.5% female) and grade level (46.3% sophomore), which is indeed difficult to completely avoid in large-scale field surveys. Therefore, during the data analysis phase, we proactively implemented two statistical measures to correct potential biases arising from these factors: First, we applied weighting to the data based on gender and grade distribution to enhance the sample's representativeness of the population. Second, in all subsequent key regression analyses and mediation effect tests, we included demographic variables such as gender, grade, region, and age as control variables in the models. This ensures that the core relationships we report—between physical exercise, self-efficacy, smartphone addiction, and negative emotions—represent net effects after controlling for these demographic background factors.

We extend our gratitude once more to the reviewers for their meticulous scrutiny, which prompted deeper reflection and clarification of our methodology. We believe the above explanations effectively address concerns regarding sampling methods and further strengthen the theoretical foundation of our study's conclusions.

---

## [Decision Letter · Decision Letter 2]

24 Nov 2025

The Impact of College Students' Physical Exercise on Negative Emotion: The Chain Mediating Role of Self-efficacy and Smartphone Addiction

PONE-D-25-38262R2

Dear Dr. Nazarudin,

We’re pleased to inform you that your manuscript has been judged scientifically suitable for publication and will be formally accepted for publication once it meets all outstanding technical requirements.

Kind regards,

Leona Cilar Budler

Academic Editor

PLOS ONE

Additional Editor Comments (optional):

There are no further comments from reviewers. All comments have been taken into account and paper has been revised.

Reviewers' comments:

Reviewer's Responses to Questions

**Comments to the Author**

Reviewer #2: All comments have been addressed

Reviewer #4: All comments have been addressed

2. Is the manuscript technically sound, and do the data support the conclusions?

Reviewer #2: Yes

Reviewer #4: Yes

3. Has the statistical analysis been performed appropriately and rigorously?

Reviewer #2: Yes

Reviewer #4: Yes

4. Have the authors made all data underlying the findings in their manuscript fully available?

Reviewer #2: Yes

Reviewer #4: Yes

5. Is the manuscript presented in an intelligible fashion and written in standard English?

Reviewer #2: Yes

Reviewer #4: Yes

Reviewer #2: Thank you very much to the editors and authors for their responses. Overall, I believe this paper meets the journal's publication requirements, and the significance of the research is also very good.

Reviewer #4: After two rounds of revisions, the quality of this article has improved a lot. I request the editorial department to publish this paper.

**Do you want your identity to be public for this peer review?** For information about this choice, including consent withdrawal, please see our Privacy Policy

Reviewer #2: No

Reviewer #4: No

---

## [Editor Report · Acceptance letter]

PONE-D-25-38262R2

PLOS One

Dear Dr. Nazarudin,

I'm pleased to inform you that your manuscript has been deemed suitable for publication in PLOS One. Congratulations! Your manuscript is now being handed over to our production team.

Kind regards,

on behalf of

Dr. Leona Cilar Budler

Academic Editor

PLOS One